# The *Photorhabdus asymbiotica* virulence cassettes deliver protein effectors directly into target eukaryotic cells

Isabella Vlisidou[1], Alexia Hapeshi[2], Joseph RJ Healey[2], Katie Smart[2], Guowei Yang[3][†]*, Nicholas R Waterfield[2][†]*

[1]All Wales Genetics Laboratory, Institute of Medical Genetics, University Hospital of Wales, Cardiff, United Kingdom; [2]Warwick Medical School, Warwick University, Coventry, United Kingdom; [3]Beijing Friendship Hospital, Capital Medical University, Beijing, China

**Abstract** *Photorhabdus* is a highly effective insect pathogen and symbiont of insecticidal nematodes. To exert its potent insecticidal effects, it elaborates a myriad of toxins and small molecule effectors. Among these, the *Photorhabdus* Virulence Cassettes (PVCs) represent an elegant self-contained delivery mechanism for diverse protein toxins. Importantly, these self-contained nanosyringes overcome host cell membrane barriers, and act independently, at a distance from the bacteria itself. In this study, we demonstrate that Pnf, a PVC needle complex associated toxin, is a Rho-GTPase, which acts via deamidation and transglutamination to disrupt the cytoskeleton. TEM and Western blots have shown a physical association between Pnf and its cognate PVC delivery mechanism. We demonstrate that for Pnf to exert its effect, translocation across the cell membrane is absolutely essential.
DOI: https://doi.org/10.7554/eLife.46259.001

*For correspondence:
yangguowei@hotmail.com (GY);
n.r.waterfield@warwick.ac.uk
(NRW)

[†]These authors contributed equally to this work

Competing interests: The authors declare that no competing interests exist.

## Introduction

Bacteria belonging to the Enterobacteriaceae genus *Photorhabdus* exist in a symbiotic partnership with entomopathogenic *Heterorhabditis* sp. nematodes. This Entomopathogenic Nematode complex (EPN) comprises a highly efficient symbiosis of pathogens that is commonly used as a biological agent to control crop pests (*Forst et al., 1997*). The *Photorhabdus* bacteria are delivered into the hemocoel of the insect, after regurgitation from the worm, where they resist the insect immune response and rapidly kill the host via septicaemic infection. Insect tissues are subsequently bio-converted into a dense soup of *Photorhabdus* bacteria, which provide a food source to support the replication of the nematode. As food resources are depleted *Photorhabdus* re-associates with infective juvenile nematodes, and together they emerge from the insect cadaver able to re-infect a new host (*Ciche et al., 2008*; *Somvanshi et al., 2012*). Three major species have been formally recognized to date within the genus - *P. luminescens*, *P. asymbiotica*, and *P. temperata*. It should be noted however that with increasing numbers of *Photorhabdus* genome sequences becoming available, the genus structure is under revision (*Machado et al., 2018*). In addition to the normal insect life cycle, *P. asymbiotica* is also the etiological agent of a serious human infection termed *Photorhabdosis*, which is associated with severe ulcerated skin lesions both at the initial infection foci and later at disseminated distal sites (*Gerrard et al., 2004*; *Gerrard et al., 2006*; *Gerrard et al., 2003a*; *Gerrard et al., 2003b*).

The *Photorhabdus* genome encodes a diverse repertoire of virulence genes encoding for protein toxins, proteases and lipases for combating diverse hosts, that can be found in chromosomally encoded pathogenicity islands (*Waterfield et al., 2009a*; *ffrench-Constant, 2007*;

**eLife digest** *Photorhabdus* are the only known group of non-marine bacteria that can produce their own light. These organisms prey on insects, which then glow in the dark once infected. The group also has an unusual weapon system formed of miniscule needle-like structures that can be sent out in the environment. These '*Photorhabdus* virulence cassettes' are loaded with toxins that are injected inside host cells; the cassettes alone can kill a caterpillar within minutes. However, it is still unclear how exactly these structures work: are they like poison darts, with the toxin on the outside, or like hypodermic needles, with the toxin within?

*Photorhabdus* bacteria make lots of deadly substances, so to look at the needles on their own, Vlisidou et al. had them produced by another species of bacteria that does not carry these weapons. The experiments revealed that the cassettes packaged the toxic proteins inside, like a hypodermic needle. Alone in the environment, the toxin cannot penetrate host cells.

Creating the cassettes takes a lot of energy, and a closer look at *Photorhabdus* showed that, at any given time during an infection, only a small number of bacteria produce them. It is therefore possible that the bacteria share the high cost of making these deadly devices by using a division of labour approach.

With a better understanding of the cassettes, it could one day become possible to harness these molecular machines to control insect pests or parasites.

DOI: https://doi.org/10.7554/eLife.46259.002

*Waterfield et al., 2002*; *Duchaud et al., 2003*; *Waterfield et al., 2009b*; *Wilkinson et al., 2009*). In addition, the bacteria also secrete a potent cocktail of other biologically active small molecules to preserve the insect cadaver in the soil from competing saprophytes and microbial predators such as amoeba (*Cai et al., 2017*; *Bozhüyük et al., 2017*). Several classes of *Photorhabdus* protein insecticidal toxins have now been well characterised including the Toxin Complexes (*ffrench-Constant and Bowen, 2000*; *Waterfield et al., 2001a*; *Waterfield et al., 2001b*; *Meusch et al., 2014*; *Erickson et al., 2007*; *Waterfield et al., 2007*; *Hares et al., 2008*), the binary PirAB toxins (*Waterfield et al., 2005*; *Ahantarig et al., 2009*; *Sirikharin et al., 2015*) and the large single polypeptide Mcf ('makes caterpillars floppy') toxins (*Daborn et al., 2002*; *Waterfield et al., 2003*; *Dowling et al., 2007*).

A fourth class of highly distinct toxin delivery systems first identified in *Photorhabdus* are the '*Photorhabdus* virulence cassettes', or PVCs (*Yang et al., 2006*). These represent operons of around 16, conserved, structural and synthetic genes (from hereon just described as the structural genes) encoding for a phage 'tailocin' like structure (*Ghequire and De Mot, 2015*) and one or more tightly linked downstream toxin-effector like genes. Genomic analysis of multiple strains of *Photorhabdus* revealed they often encode up to five or six copies of the operon, each with unique downstream effector genes (*Hapeshi and Waterfield, 2017*).

It should be noted that PVC-like elements are not restricted to *Photorhabdus* as a well-characterized homologous operon can also be found on the pADAP plasmid of the insect pathogenic bacteria *Serratia entomophila* (*Hurst et al., 2004*). This system has been named the anti-feeding prophage (AFP), as it is responsible for the cessation of feeding in the New Zealand grass grub host. Recent cryo-electron microscopy studies have revealed that, morphologically, AFP resembles a simplified version of the sheathed tail of bacteriophages such as T4, including a baseplate complex. It also shares features with type-VI secretion systems, with the central tube of the structure having a similar diameter and axial width to the Hcp1 hexamer of *P. aeruginosa* T6SS (*Heymann et al., 2013*). One important difference between the PVC and T6SS machinery is that the T6SS relies upon direct contact between host and bacterial cell, and is anchored in to the membrane by a substantial membrane complex whose structure is still being elucidated (*Durand et al., 2015*), whereas the PVC needle complex is freely released into the surrounding milieu and so can act at a distance.

Furthermore, recent reports have indicated that other more diverse bacteria can also make similar needle complexes for manipulation of eukaryotic hosts. A well-characterized example is the production of analogous devices by the marine bacterium *Pseudoalteromonas luteoviolacea* (*Figure 1A*). These structures are involved in the developmental metamorphosis of the larvae of the tubeworm

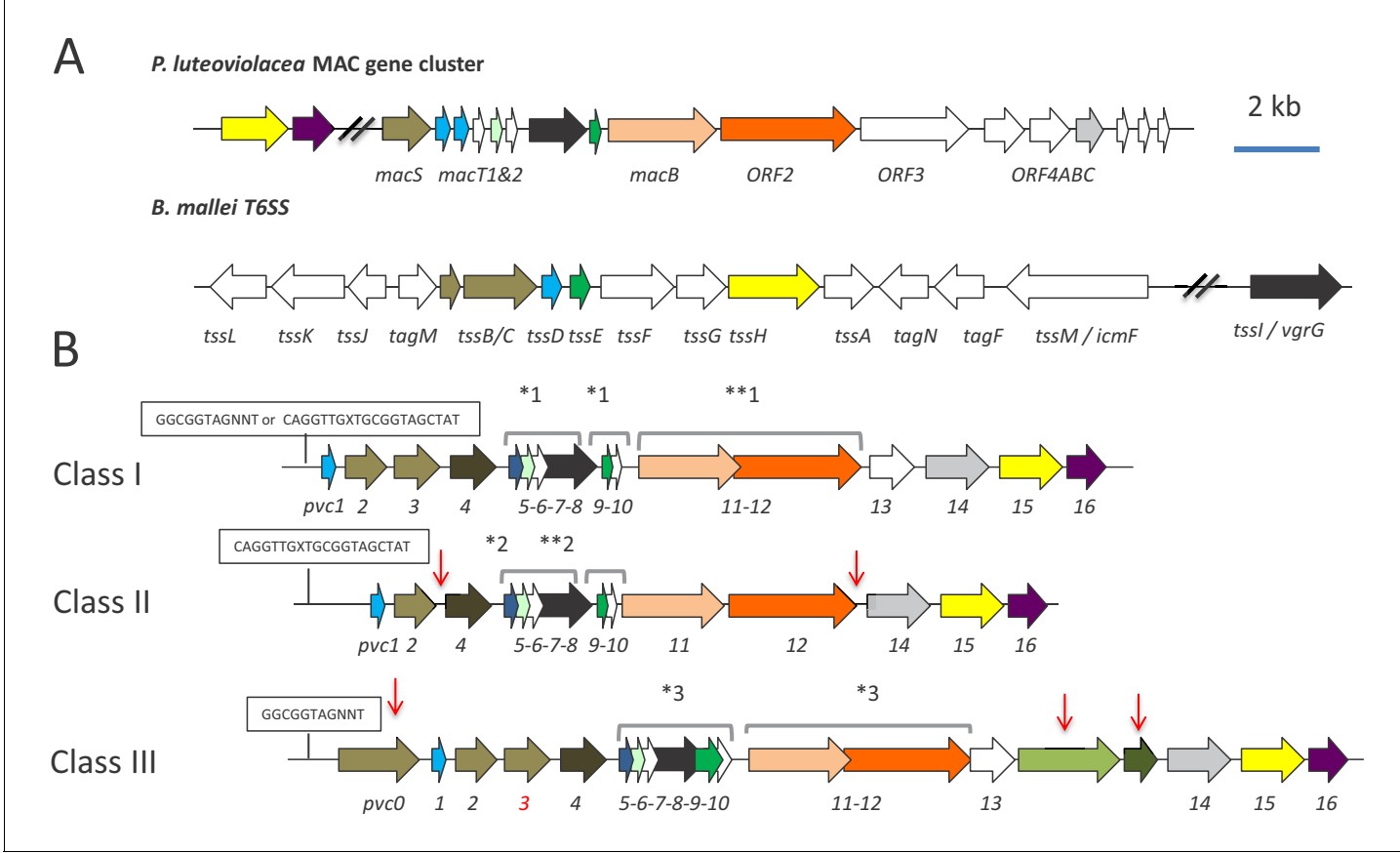

**Figure 1.** Genetic organisation of PVC operons. (**A**) Similarity between PVCs and two diverse protein secretion systems, (i) the *P. luteoviolacea mac* gene cluster and (ii) The type-VI secretion system (T6SS) from *Burkholderia mallei*. Homologous protein sequences are coloured coded. (**B**) Three classes of PVC structural operons observed in the genomes of *Photorhabdus* and members of other genera. Classes 1–3 are exemplified by PVC*pnf*, PVC*lopT* and PVC*PaTox* respectively. Homologous genes are colour coded. Red vertical arrows represent variations relative to the representative class I PVC*pnf* operon of *P. asymbiotica* ATCC43949 (genes *pvc1*-PAU_03353 to *pvc16*-PAU_03338). Predicted functions of individual Pvc proteins based on homology to known proteins can be seen in *Figure 1—figure supplement 1*. The boxed 'GGCGGTAGNNT' or 'CAGGTTGXTGCGGTAGCTAT' sequences represent positions of the conserved RfaH anti-termination protein and cryptic operator sequences respectively. Square brackets above certain genes indicate apparent translational coupling. More specifically; *one indicates coupling in PVC*pnf* and PVC*cif* of *Pl*[TT01], *Pa*[ATCC43949], *Pa*[PB68] and *Pa*[Kingscliff] and in the *Serratia entomophila afp* operon in addition to an uncharacterised PVC in *Yersinia ruckeri* ATCC 29473. **one indicates these genes are not coupled in *Pa*[Kingscliff]. *two indicates coupling in PVC*lopT* of *Pl*[TT01], *Pa*[ATCC43949], *Pa*[PB68] and *Pa*[Kingscliff]. **two indicates these genes are not coupled in *Pa*[Kingscliff]. *three indicates coupling in PVC*Patox* of *Pa*[ATCC43949] and *Pa*[Kingscliff] (although *pvc11* possibly contains a frame-shift in *Pa*[Kingscliff]). The *pvc3* is also deleted in *Pa*[Kingscliff].

DOI: https://doi.org/10.7554/eLife.46259.003

The following figure supplements are available for figure 1:

**Figure supplement 1.** PVC operons in relation to putative regions of horizontal gene transfer as identified by Alien Hunter.
DOI: https://doi.org/10.7554/eLife.46259.004

**Figure supplement 2.** Boxplots of the mean GC content across 16 different *pvc* operons of *Photorhabdus*.
DOI: https://doi.org/10.7554/eLife.46259.005

**Figure supplement 3.** Box plots of amino acid similarity across homologous protein sequences for these same 16 operons.
DOI: https://doi.org/10.7554/eLife.46259.006

**Figure supplement 4.** A map of the model class I *Pa*[ATCC43949] PVC*pnf* operon showing two effector genes in the payload region in red and a table, both colour matched to the description of subunit-structure relationship described in the recent CryoEM structure paper (*Jiang et al., 2019*).
DOI: https://doi.org/10.7554/eLife.46259.007

*Hydroides elegans*, and they are deployed in outward-facing arrays comprising about 100 contractile structures, with baseplates linked by tail fibres in a hexagonal net (*Shikuma et al., 2014*). Interrogation of sequence databases with PVC protein sequences suggests many other more diverse tailocin-like systems are yet to be characterized (*Sarris et al., 2014*). These include operons closely

related to the PVCs in *Xenorhabdus bovienii* CS03, *Yersinia ruckeri* ATCC29473 and *Vibrio campbellii* AND4. In addition, evidence of more diverse elements, like that of *P. luteoviolacea*, can also be seen. To address this, we have recently performed an exhaustive analysis of all available prokaryotic and archaeal genome sequences in the public databases to look at the distribution of *pvc*-like elements (unpublished data). This suggests that PVC-like nano-syringes and their distant cousins are of enormous ecological and perhaps biomedical significance.

Here we focus on a single *Photorhabdus pvc* operon (which elaborates the PVC*pnf* needle complex (*Yang et al., 2006*) to understand the relationship between the structural genes and the tightly linked effector gene, *pnf*. We confirm in vivo expression during insect infection and reveal a high level of population heterogeneity of expression in vitro. We demonstrate for the first time the physical association of the Pnf effector toxin protein with the secreted structural needle complex using Western blot and electron microscopy. Furthermore, we prove that the cognate Pnf effector needs to be delivered into the eukaryote cell cytoplasm to exert any measurable effect and confirm its predicted activity targeting small Rho-GTPase target proteins. Taken together this work describes an important new class of protein toxin secretion and injection delivery systems which, unlike the well-described Types III, IV and VI systems, can act 'at a distance', requiring no intimate contact between bacteria and host cells.

## Results

### A bioinformatic analysis of *pvc* structural operon sequences

A comparison of *pvc* structural operons identified in the genome sequences of *Photorhabdus* and certain members of other genera, available at the time of publication (*Duchaud et al., 2003*; *Wilkinson et al., 2009*; *Wilkinson et al., 2010* and our unpublished data), allowed us to define three distinct genetic sub-types. The PVC*pnf* operon belongs to class I, which has 16 structural genes and three translationally coupled gene blocks, and is of the type typically seen in non-*Photorhabdus* genera. Class II and III operons differ in the number of structural genes and translationally coupled gene blocks (*Figure 1B*). Given the diversity of *pvc*-operons, and their typically poor annotation in genome sequences, it is necessary here to define a nomenclature protocol to allow reference to any given operon. An example of the method we have adopted is as follows; [*Pa* ^ATCC43949^ PVC*pnf*], where *Pa* ^ATCC43949^ is species and strain, in this case *Photorhabdus asymbiotica* strain ATCC43949 and PVC*pnf* is the specific operon within that genome with the suffix referring to one of the tightly linked effectors, in this case the *pnf* effector gene. We will also include gene identifiers for either end of the operon where appropriate, which in this case would be PAU_03353-PAU_03332, which are the genes for *pvc1* and *pnf* respectively.

With reference to published literature and a detailed bioinformatic analysis of promoter regions upstream of the *pvc1* genes, we can identify two distinct, potential *cis*-operator sequences. Firstly operons belonging to classes I (e.g. PVC*pnf*) and III (e.g. PVC*PaTox*) typically encode the highly conserved RfaH operator sequence, GGCGGTAGNNT (*Belogurov et al., 2009*). RfaH is a conserved anti-termination protein that is known to regulate large operons encoding for extracellular factors in *E. coli*. It is also believed to be important in ensuring appropriate transcriptional control of horizontally acquired operons (*Belogurov et al., 2009*). Many of the *pvc*-operons in *Photorhabdus* and members of other genera (including *Xenorhabdus* and *Yersinia*) encode this operator sequence. An unusual example is the pADAP plasmid encoded *Serratia entomophila* anti feeding prophage (*afp*) (*Hurst et al., 2004*). While the *afp* promoter also encodes an RfaH operator sequence, it has been demonstrated that it is positively regulated by a tightly linked specific regulator protein, AnfA1 (*Hurst et al., 2007*; *Hurst et al., 2003*). This protein is a distant homologue of RfaH suggesting that other class I or III *pvc* operons are not necessarily under the regulation of the chromosomal RfaH orthologue, but might also be controlled by other diverse regulators that utilise this same operator sequence (*Carter et al., 2004*). Unfortunately, we were not able to directly test the role of RfaH in *pvc*-regulation as attempts to knock out the single *P. luminescens* TT01 *rfaH* gene were not successful, suggesting an essential role. Secondly, all class II operons (e.g. PVC*lopT*) and certain class I operons (e.g. PVC*units*1-4) encode a minimal cryptic conserved sequence motif, CAGGTTGXTGCGGTAGCTAT. In both cases these conserved *cis*-encoded sequences are located between the *pvc1* gene and the transcription start sites, as defined by previous RNA-seq analysis (*Mulley et al., 2015*

and unpublished data). We speculate that this motif, which we have so far only found in *pvc* loci, may represent an operator sequence for a cryptic regulator.

Although there is no experimental evidence, several observations suggest that horizontal gene transfer may have been responsible for the dissemination of many observed *pvc*-operons. These include; the presence of four *pvc* operons in tandem in *P. luminescens* TT01 (directly adjacent to a type IV DNA conjugation pilus operon), the presence of multiple *pvc* operons in any given genome, the suggestion that several operons are regulated by RfaH and the remnants of YhgA-like IS elements and inverted repeats (IR) associated with many of the operons (e.g. the IR sequence 5'-TTATA TTGAA(t/g)GAATATTAAGCAAGAAAC-3' flanking [$Pl^{TT01}$PVC*u4*]). Nevertheless, automatic prediction of horizontal gene transfer regions (HGTs) using Alien Hunter 1.7 (*Vernikos and Parkhill, 2006*) either did not detect any HGT elements spanning the structural regions of PVCs or in the cases where such an element was detected it was assigned a low confidence score (*Figure 1—figure supplement 1*).

An analysis of the conservation of individual genes across different *pvc* operons at both DNA and protein sequence levels suggests that either recombination or diversifying selection is more likely to have occurred in the more 3' regions of the operons (*Figure 1—figure supplement 2*). This is perhaps no surprise as each *pvc* operon can be seen to encode different effector genes in the 3' payload region of the operons. An analysis of conservation of protein sequences of the *pvc* operons showed that within *pvc*-operons a good deal of variability is possible while presumably retaining the ability to produce a similar macromolecular structure (*Figure 1—figure supplement 3*). This is supported by HHPRED structural homology comparisons for equivalent PVC proteins across different operons, despite often-variable primary amino acid sequences (data not shown). We note that the most diverse protein seen in *pvc*-operons is that of the predicted tail fibre proteins, Pvc13, which we may expect if different *pvc*-operons are adapted for different host cell targets. Paralogous genes within *pvc*-operons include *pvc1* and *pvc5* which encode homologs of Hcp, the inner tube protein of contractile tube mechanisms such as T6SS and phage protein Gp27 and *pvc2*, *−3* and *−4* which encode homologues of the outer sheath proteins of phage (*Leiman et al., 2010*) and T6SS (*Russell et al., 2014*). *Figure 1—figure supplement 4* illustrates the organisation of the [$Pa^{ATCC43949}$ PVC*pnf*] operon used as a model system in our experimental studies described here, showing the top HHPRED structural homology hits and predicted roles for each encoded protein at the time of writing.

## A bioinformatic analysis of *pvc*-operon effector gene sequences

A comparison of the 3' effector 'payload regions' of different *pvc* operons reveals a large diversity of effector genes, with a range of predicted activities, covering a large range of sizes and isoelectric point values (data not shown). Some operons encode only a single putative effector, for example [$Pa^{ATCC43949}$ PVC*PaTox* PAU_02249–02230] while others have several, either tandem homologues of one another, for example [$Pa^{ATCC43949}$ PVC*u4* PAU_02790–02808] or entirely unrelated putative effector genes, for example [$Pa^{ATCC43949}$ PVC*lopT* PAU_02112–02095]. Many effector genes are also tightly linked to transposase gene remnants suggesting they are typically exchanged by horizontal acquisition. This is further supported by the observation that orthologous *pvc*-operons in the same chromosomal context may have different effector genes in different strains. A good example of this being the unrelated effector genes seen in the orthologous structural 'PVC*pnf*' operon loci of $Pa^{Kingscliff}$ and $Pa^{ATCC43949}$ which carry a tyrosine glycosylase and Pnf (this paper) respectively. Analysis with Alien Hunter 1.7, suggests that certain *pvc*-operon/effector associations are ancestral to any given species. For example, the association of the *pvc17* effector with PVC*u4*, and the multiple linked effectors with the PVC*lopT* operon in both $Pa^{ATCC43949}$ and $Pl^{TT01}$. Conversely other *pvc*-operons show evidence of recent horizontal acquisition of their 3'linked effectors, for example PVC*cif* and PVC*pnf* (not shown).

## Expression of PVC*pnf* in vitro and in vivo

A previous RNA-seq analysis of global transcription in three strains; *P. asymbiotica*$^{ATCC43949}$ (*Mulley et al., 2015*), *P. asymbiotica*$^{Kingscliff}$ and *P. luminescens*$^{TT01}$ (unpublished) showed condition dependent expression of certain *pvc*-operons but not all. Therefore, due to the diversity of *pvc* operons and effectors in *Photorhabdus*, we focused on a single model class I *pvc* operon, [$Pa^{ATCC43949}$

PVC*pnf*], to elucidate the relationship between the conserved structural and effector proteins. This operon was selected as it elaborates a well-defined needle complex structure (as observed by electron microscopy) which has potent insect killing activity when heterologously expressed in *E. coli* (*Yang et al., 2006*). This operon has two putative effector genes in the downstream 'payload region', PAU_03337, which shows similarity to adenylate cyclase toxins (e.g. the anthrax Edema Factor and Pseudomonas ExoY toxin) and *pnf* (PAU_03332). While the predicted activity of PAU_03337 has not been tested directly, when expressed in the NIH-3T3 cell cytoplasm (in transient transfection experiments) it did produce a highly unusual cytoskeleton phenotype (*Yang et al., 2006*). Pnf (Photorhabdus necrosis factor) is a homologue of the active site domain of the *Yersinia* CNF2 (Cyto Necrosis Factor 2) toxin, which has small-GTPase deamidase and transglutaminase activities (*Knust and Schmidt, 2010*).

In order to confirm the expression of this model *pvc*-operon in *Photorhabdus* during an insect infection we constructed transcription-translation reporter plasmids in which the promoter regions and the first 150 bp of coding sequence of *pvc1*, *pnf* [both from $Pa^{ATCC43949}$ PVC*pnf*] and the *P. asymbiotica* chromosomal *rpsM* ribosomal 'housekeeping' gene (as a positive control) were genetically fused in frame to a *gfpmut2* gene with no start codon (referred to hereon as *pvc1::gfp*, *pnf::gfp* and *rpsM::gfp* reporters). Note, the genomic context and our previous unpublished RT-PCR studies suggested that *pnf* had its own promoter and could be transcribed independently of the *pvc* structural genes. As we are unable to transform $Pa^{ATCC43949}$ itself, these plasmids were transformed into the well-characterised and genetically tractable strain *P. luminescens*$^{TT01}$ to provide suitable reporter strains for in vitro and in vivo expression studies. For in vitro studies we cultured the bacteria in LB medium supplemented with *Manduca sexta* clarified hemolymph and grown to late stationary phase, before microscopic examination (*Figure 2A*). For in vivo studies, we injected a low inoculum (c.a. 100 CFU) of the reporter strains into *M. sexta,* and allowed the infection to establish for 12 hr before macroscopic examination of insect tissues in situ using a (fluorescence) dissecting microscope. We also took hemolymph samples from these insects and visualised the hemocytes and bacteria microscopically using confocal microscopy. This approach provides confidence that the cells showing *gfp* expression are those in the process of a normal productive infection rather than those carried through in the inoculum.

*Figure 2A* shows expression of GFP reporter from the *rpsM* positive control and both the *pvc1::gfp* and *pnf::gfp* reporters in LB supplemented with *M. sexta* hemolymph, although not in all cells of the bacterial population (see below). Furthermore, we also saw expression in bacteria in the ex vivo hemolymph samples taken during infection of live insects (*Figure 2A*). It was also possible to confirm expression of *pnf::gfp* in bacteria attached to the insect trachea in localised putative biofilm masses. In this case, while the expected insect melanisation immune response could be seen to have occurred elsewhere on the trachea, it was notably absent from the *pnf* expressing bacterial biomass (*Figure 2B*).

We subsequently expanded this analysis to include a panel of transcription-translation reporter plasmids for different PVC operons from two different species of *Photorhabdus*. We cloned around 500 bp of the 5' regions (containing the promoter, translation initiation region and first ATG of *pvc1* translationally fused to *gfp*) of *P. luminescens*$^{TT01}$; PVC*unit1*, PVC*unit4*, PVC*lopT* and PVC*cif* and from *P. asymbiotica* $^{PB68.1}$; PVC*pnf*, PVC*unit*, PVC*lopT* and PVC*cif* (see supplementary file 'Reporter construct primers'). Each of the eight reporter constructs were then transformed into the relevant *Photorhabdus* strain and examined using fluorescent microscopy to assess the expression patterns across growth phases, when grown in LB with aeration and maintaining plasmid marker selection. We initially focused on the *P. asymbiotica* $^{PB68.1}$ [PVC*pnf*] reporter strain as it represents an orthologue of the *P. asymbiotica* $^{ATCC43949}$ PVC*pnf* operon, which is the model operon central to this report. We observed a high level of population heterogeneity in expression, usually with only very few cells expressing GFP at any one time (*Figure 2—figure supplement 1*). Image analysis provides an objective assessment of heterogeneity, and small percentage of cells expressing Gfp, and thus the *pvc*-operon (*Figure 2—figure supplement 2*). Interestingly, a similar pattern of heterogeneity in expression was seen for the other seven operon reporter constructs from both $Pa^{PB68}$ and $Pl^{TT01}$ (*Figure 2—figure supplements 3* and *4*). Note in certain cases we saw no expression at all (e.g. the PVC*lopT* operon of $Pa^{PB68.1}$), which fits with the low or lack of mRNA expression seen in a previous in vitro RNAseq study for the majority of *pvc*-operons (upblished data). We also assessed

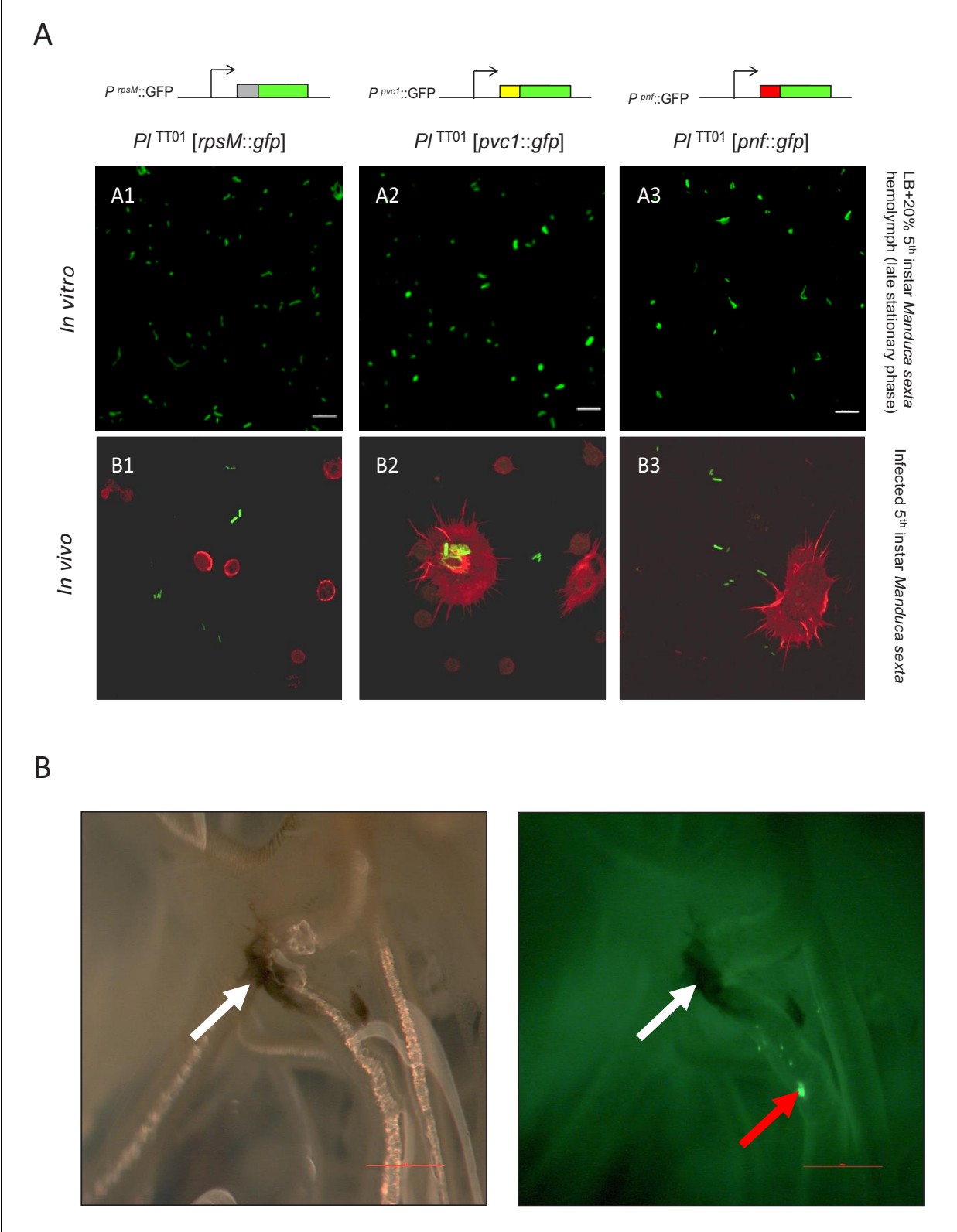

**Figure 2.** Examination of PVC operon expression using *gfp* reporter constructs in *Photorhabdus*. (**A**) Confocal micrographs of showing in vitro (**A1–A3**) and in vivo (**B1–B3**) expression in *Pl*^TT01 of *gfp* transcription-translation reporter constructs of *Pa*^ATCC43949 PVC*pnf* operon gene promoters. These plasmid-based reporters were constructed by fusing the transcription promoter regions and the first 37 codons of the target gene in frame with the second codon of *gfp*. Target gene promoters shown are (**A1 and B1**) the constitutively expressed *rpsM* gene, (**A2 and B2**) the *Pa*^ATCC43949 PVC*pnf pvc1*

*Figure 2 continued on next page*

*Figure 2 continued*

structural gene and (**A3 and B3**) $Pa^{ATCC43949}$ PVC*pnf pnf* payload toxin gene. The in vitro panels (A1-3) show reporter expression after growth in LB supplemented with 20% (v/v) 5th instar *M. sexta* hemolymph at late stationary phase. The in vivo panels show ex vivo hemolymph from 5th instar *M. sexta* infected with $Pl^{TT01}$ harbouring the three different reporter constructs. The hemocyte cytoskeletons are stained red with TRITC-Phalloidin conjugate. (**B**) White light (left) and fluorescence illumination (right) of the trachea of a dissected 5th instar *M. sexta* previously infected with $Pl^{TT01}$ harbouring the $Pa^{ATCC43949}$ PVC*pnf pnf::gfp* reporter construct. Brightly fluorescent green bacteria were detected in association with the trachea (red arrow) in close proximity to melanotic nodules (white arrows) at 12 hr post infection, demonstrating the induction of the *pnf* promoter and the production of the Pnf::GFP fusion in situ. Bars show 0.1 mm.

DOI: https://doi.org/10.7554/eLife.46259.008

The following source data and figure supplements are available for figure 2:

**Figure supplement 1.** A representative selection of images for four time points, for *P. asymbiotica* PB68.1 ($Pa^{PB68}$) harbouring.

DOI: https://doi.org/10.7554/eLife.46259.009

**Figure supplement 2.** Quantification of population heterogeneity of PVC expression using image analysis of *gfp* expression.

DOI: https://doi.org/10.7554/eLife.46259.010

**Figure supplement 2—source data 1.** Cell intensity counts used in image analysis quantification shown in *Figure 2—figure supplement 2*.

DOI: https://doi.org/10.7554/eLife.46259.011

**Figure supplement 3.** A representative selection of images for four growth time points, for *P. asymbiotica*.

DOI: https://doi.org/10.7554/eLife.46259.012

**Figure supplement 4.** A representative selection of images for four growth time points, for *P. luminescens*.

DOI: https://doi.org/10.7554/eLife.46259.013

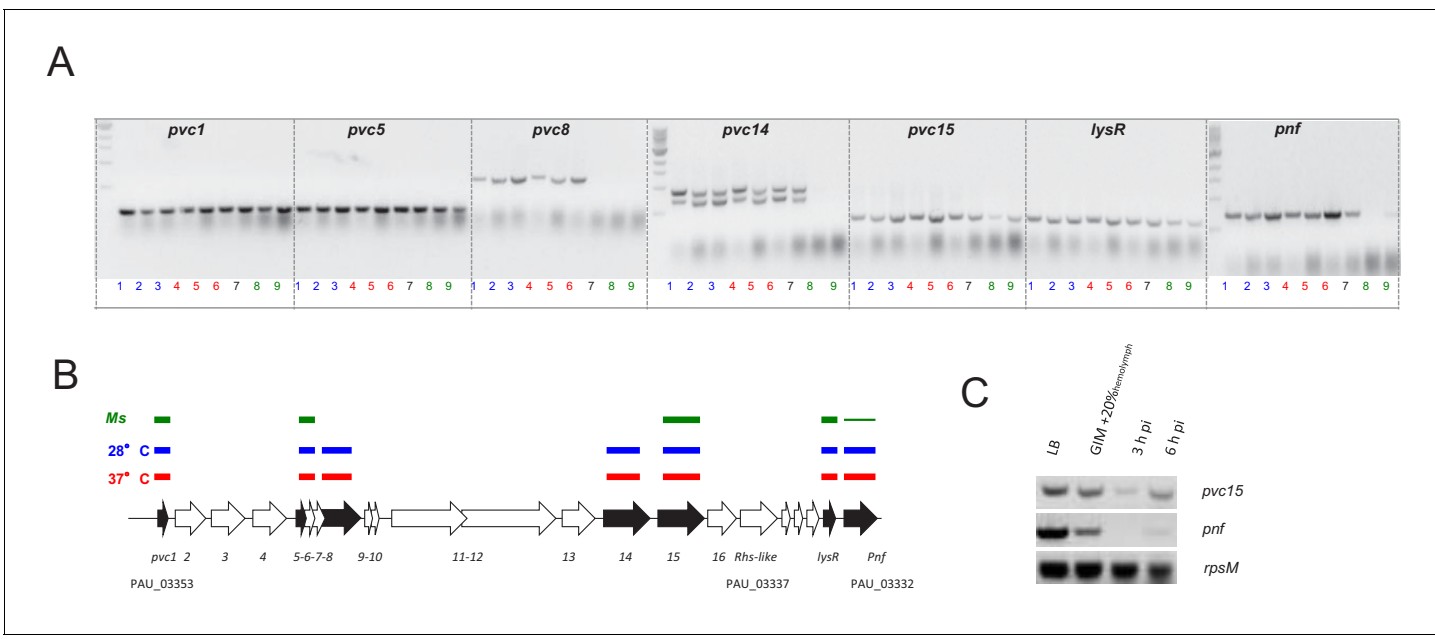

**Figure 3.** Confirmation of the PVCpnf-operon transcription in vitro and in vivo using RT-PCR. (**A**) RT-PCR analysis of gene transcription of various genes of the $Pa^{ATCC43949}$ PVC*pnf* operon over time in vitro at insect (28°C) and human (37°C) relevant temperatures and in vivo during *Manduca sexta* (*Ms*) infection. Lane key; lanes 1, 2 and 3 (blue) represent in vitro growth in aerated LB at 28°C for 4, 8 and 24 hr respectively; lanes 4, 5 and 6 (red) are growth in aerated LB at 37°C for 4, 8 and 24 hr; lane 7 (black) is growth in LB at 28°C for 16 hr; lanes 8 and 9 (green) are from 3 hr and 6 hr post infection blood of *Ms* infected with *P. asymbiotica* at 28°C. (**B**) Map of the operon showing RT-PCR target genes in black. The lane-colour coded bars above the ORFs summarise in which conditions gene transcription could be detected. Note *pvc8* and *pvc14* mRNA could not be detected from infected *Ms* and the *pnf* mRNA was only detected after 6 hr of infection. (**C**) RT-PCR signals for *pvc15* and *pnf* from infected insects with the *rpsM* (ribosomal subunit protein S13) loading control. Lanes represent (in order); 4 hr growth in LB at 28°C; 4 hr growth in Grace's insect medium supplemented with 20% (v/v) *Ms* hemolymph; 3 hr and 6 hr post infection ex vivo blood of *Ms* infected with *P. asymbiotica* at 28°C.

DOI: https://doi.org/10.7554/eLife.46259.014

expression in biofilms grown statically on glass slides and observed the same pattern, though with even fewer cells seen to express GFP (not shown).

In order to corroborate the observations made using the plasmid-based reporter constructs in *P. luminescens* [TT01] during infection we also performed RT-PCR analysis of transcription of the PVC*pnf* chromosomal operon in the original *Pa*[ATCC43939] strain. This confirmed transcription across the operon in vitro when the bacteria were grown at either 28°C or 37°C, although transcription of certain genes was difficult to detect in vivo during *Manduca sexta* infections (*Figure 3*).

## The Pnf effector protein is physically associated with the PVC needle complex

We investigated if the Pnf effector protein actually becomes physically associated with the *pvc*-encoded needle complex we had previously visualised by electron microscopy (*Yang et al., 2006*). While a recent publication has described a cryo-EM structure of this particular PVC needle complex (*Jiang et al., 2019*), they did not include expression of an effector. In order to investigate if the complex does indeed physically associate with a cognate effector, we raised anti-peptide antibodies against synthetic peptides representing amino acids 206–219 of effector Pnf (TGQKPGNNEWKTGR) and amino acids 130–143 (DGPETELTINGAEE) of the predicted outer sheath protein Pvc2. We then used these antibodies to probe PVC*pnf* needles expressed from cosmid clones in *E. coli*. Previously we used 2D-SDS PAGE analysis of PVC*pnf* needle complex produced by these same cosmid clones to confirm the presence of Pvc2, along with Pvc1, 3, 5, 11, 14 and 16 proteins (*Yang et al., 2006* and unpublished data). We confirmed specificity of the Pnf antibody using western blot analysis of extracts of *E. coli* heterologously expressing Pnf alone.

We first used the anti-Pnf peptide antibody to test for the presence of Pnf protein in supernatants from the native bacterial strain *Pa*[ATCC43949]. We initially tested for the presence of Pnf in clarified supernatants and particulate preparations of stationary phase cultures. Previous studies reporting successful detection of *Photorhabdus* secreted insecticidal toxins under similar conditions.

We could detect Pnf in preparations enriched for the complexes but not in clarified supernatants. More specifically, the Pnf protein could only be detected in the needle complex fraction of wild-type *Pa*[ATCC43949] which was isolated from 250 ml overnight cultures using DEAE-Sepharose chromatography but not in concentrated supernatant collected from a culture grown from the same starter and under the same conditions. The toxin could only be detected if the needle complexes were first either chemically or physically disrupted before electrophoresis (*Figure 4B*). Taken together these findings are consistent with the hypothesis that the Pnf protein is sequestered inside the needle complex or in some other configuration such that the TGQKPGNNEWKTGR epitope is physically hidden from access by the antibody. There is still the potential that the toxin is released in the supernatant but at not detectable amounts in these conditions.

Secondly, we enriched needle complexes from insect toxic supernatants of an *E. coli* cosmid clone that encodes the *Pa*[ATCC43949] PVC*pnf* operon, as previously described (*Yang et al., 2006*). The anti-Pnf antibody was used for in situ labelling of Pnf on Transmission Electron Microscopy grids, visualised with negative staining and an anti-rabbit gold-conjugate secondary antibody. It was only possible to detect Pnf protein near the ends of either contracted or damaged needle complexes (*Figure 4A*). Note we saw no non-specific labelling when the gold-conjugate secondary antibody was used alone. As the full cryo-EM structure of this model PVC*pnf* was recently published, it allowed us to examine the location of the epitope we used to raise and anti-peptide antibody to the main outer sheath component Pvc2. While the epitope should indeed be surface exposed based on our analysis, we nevertheless only saw an anti-Pvc2 signal associated with what appeared to be disrupted fragments of needle complexes. We therefore surmise that the secondary structure of Pvc2 in the intact needle abolishes antibody binding to the epitope we selected.

## The Pnf protein requires delivery into the eukaryotic cell cytoplasm to exert its effect

In a previous publication we reported that injection of an enriched *Pa*[ATCC43949] PVC*pnf* needle complex preparation; heterologously produced by an *E. coli* cosmid clone, caused melanisation and death of *Galleria mellonella* larvae within 30 min. In addition, microscopic analysis of phalloidin stained hemocytes taken from these dying animals revealed the cells were shrunken with highly

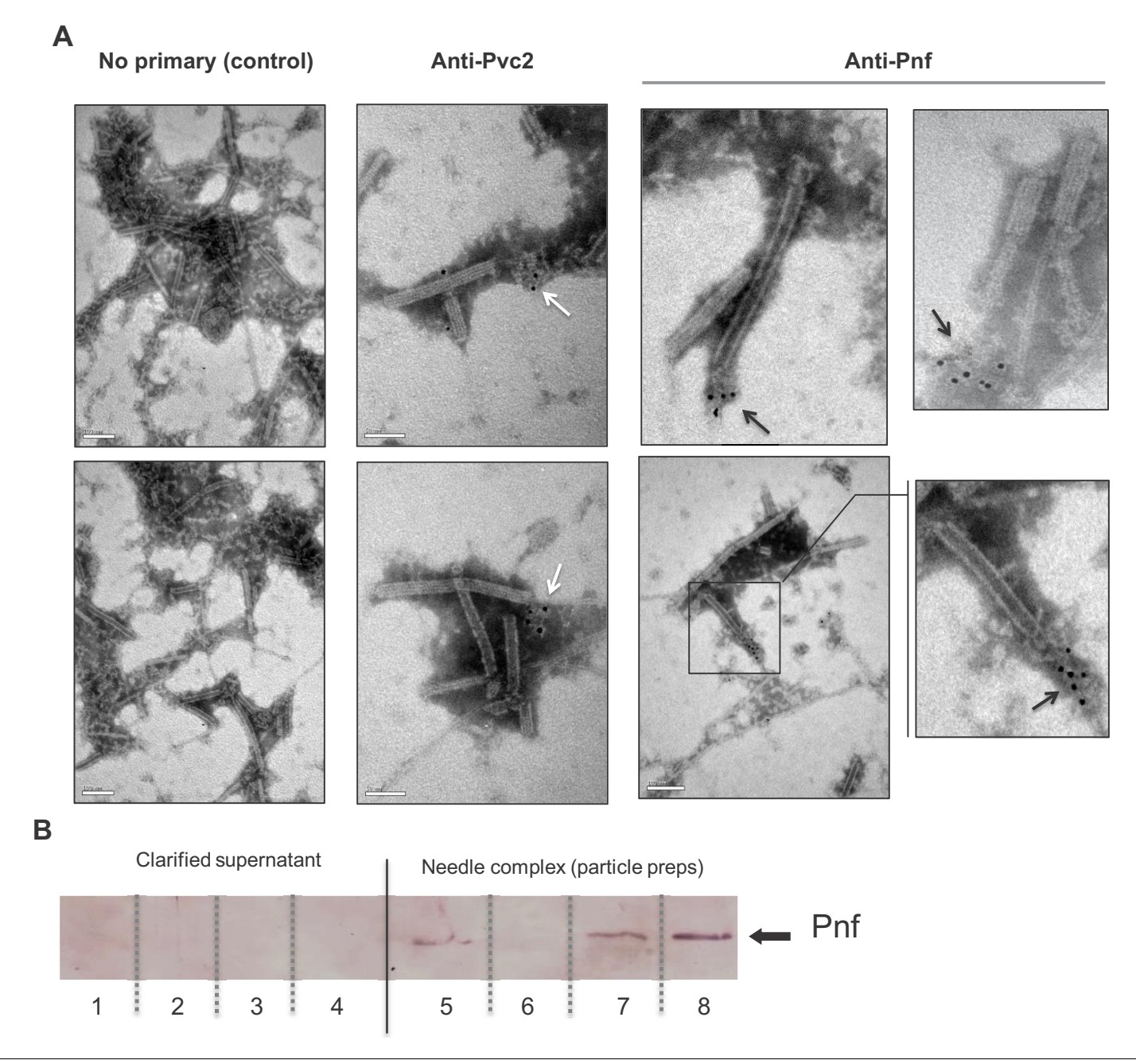

**Figure 4.** Pnf protein can only be detected in discharged or disrupted needle complexes using antibody based analysis. (**A**) Representative images of immuno-gold stained transmission electron microscopy grids confirming the Pnf-payload toxin is associated with the needle complex. PVC*pnf* needle complexes (PVC-NC) were prepared from supernatants of the *E. coli* 4df10 cosmid clone, which encodes the PVC*pnf* operon. We used anti-peptide antibodies against Pvc2 (DGPETELTINGAEE) and Pnf (TGQKPGNNEWKTGR) epitopes to localise these protein subunits. The Pvc2 epitope appeared to only become accessible to the antibody when subunits were 'broken off' the ends (white arrows). The Pnf toxin could also only be detected at the ends of broken or contracted suggesting they are contained within the complex (black arrows). (**B**) Western blot analysis confirms that the Pnf protein can only be detected using the anti-peptide antibody if the needle complex is either chemically or physically disrupted. These preparations were taken from $Pa^{ATCC43949}$ supernatants. The inability to detect Pnf in clarified supernatants confirms all the protein is associated with the PVC-NC enrichment preparation. Lanes 1 + 5; sonicated samples, 2 + 6; 1M NaCl treatments, 3 + 7; 1% SDS treatments 4 + 8; 1M Urea treatments. Note the PVC-NC appears stable in 1M NaCl.

DOI: https://doi.org/10.7554/eLife.46259.015

condensed cytoskeletons, and likely already dead. This effect was abolished by heat denaturing the preparation. In this same publication (*Yang et al., 2006*) we demonstrated that transient cytoplasmic expression of the Pnf protein caused extensive cytoskeleton re-arrangement and likely cell death in cultured Human HeLa ATCC CCL2 cells, similar to that observed in the ex vivo G. mellonella hemocytes. In an attempt to directly visualise the interaction of the heterologously produced PVC*pnf* needle complex with insect hemocytes and to determine the initial effects on the cellular morphology, we injected intact or heat denatured PVC*pnf* needle complex preparations into 5$^{th}$ instar *Manduca sexta* larvae before bleeding the animals and preparing their circulating hemocytes for surface examination by cryo-SEM. The surface of hemocytes taken from insects that had been injected with intact complex showed membrane ruffling consistent with the predicted mode of action of the Pnf protein (see below). Furthermore, we could also see linear structures approximately 150 nm in length on the surface of the cells near the sites of membrane ruffles consistent with attached needle complexes. The surface of the control hemocytes taken from insects that had been injected with heat-denatured complex remained relatively smooth and homogeneous and we saw no equivalent linear structures. *Figure 5* shows representative images from these experiments.

We wished to know if the Pnf effector could exert this toxic effect independently of the needle complex, when applied externally to eukaryotic cells. Therefore, we heterologously expressed (in *E. coli*) and purified the Pnf protein in addition to a predicted toxoid derivative. The wild-type Pnf protein and toxoid derivative were purified using HisTrap Ni$^{2+}$-affinity columns with the fast phase liquid chromatography (FPLC) AKTA system as described in the methods. SDS-PAGE was used to confirm high levels of purity and that no obvious degradation had occurred. The toxoid was designed based on homology between Pnf and the CNF2 toxin active site, wherein we mutated the cysteine at amino acid position 190 into an alanine (Pnf $^{C190A}$). Firstly, neither purified wild type nor toxoid proteins had any obvious toxic effect when injected into cohorts of *G. mellonella*, even at high doses (data not shown). We subsequently used bioPORTER, a liposome based transfection system, to introduce the purified proteins directly into cultured human cells. We visualised effects on the cytoskeleton and nucleus using TRITC-phalloidin and DAPI staining respectively. The wild type Pnf protein had a very clear effect on the cells, producing phenotypes consistent with those predicted by similarity to the CNF2 toxin. CNF2 is known to modify the cellular Rho GTPases, RhoA, Rac1 and Cdc42. Pnf delivery as a bioPORTER formulation led to the formation of F-actin filaments within 24

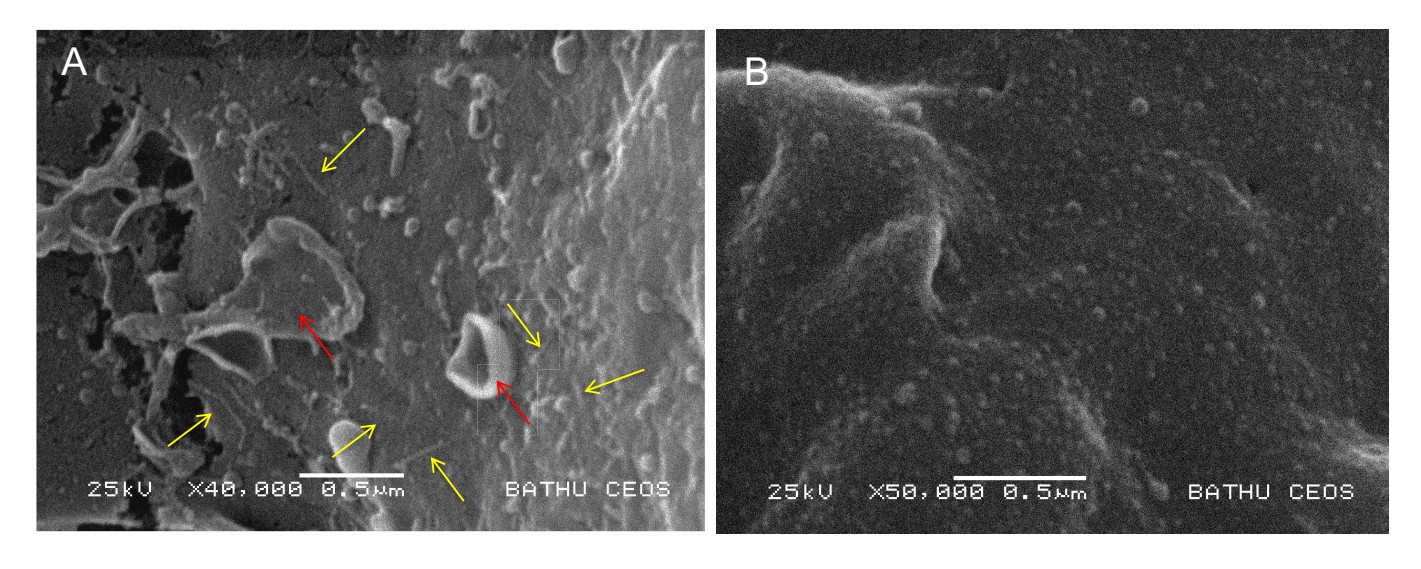

**Figure 5.** Visualisation of PVC needle complexes on the cell surface. Cryo-SEM analysis of ex vivo hemocytes from 5$^{th}$ instar *Manduca sexta* that had been injected with (**A**) native or (**B**) heat inactivated enriched preparations of *Pa*$^{ATCC43949}$ PVC*pnf* needle complexes heterologously produced by the *E. coli* cosmid clone. Note the abundant linear structures believed to be the PVC needle complex (yellow arrows) and membrane ruffling effect (red arrows) absent from the control treatment.
DOI: https://doi.org/10.7554/eLife.46259.016

hr followed by multi-nucleation by 48 hr, phenotypes consistent with the modification of the Rho GTPases. The toxoid derivative, delivered at the same dose using the same approach, produced no changes, giving cellular phenotypes consistent with that of the negative control or of the wild-type Pnf protein topically applied without the bioPORTER transfection agent (*Figure 6*). Computational secondary structure predictions of the toxin and toxoid were performed to assess the potential impact of the C190A active site amino acid substitution using the Phyre2 algorithm. We obtained structural predictions (100% confidence score) based on similarity to the known structure of a *E. coli* CNF1/2 family toxin active site domain (PDB Entry: 1hq0), which revealed only very minor predicted changes (*Figure 6—figure supplement 1*).

## The pnf protein effector modifies small rho GTPases

Based on homology to CNF2 the effect of Pnf on target cell proteins is predicted to include the modification of several Rho-family GTPases. Therefore, we used western blot assays to examine in

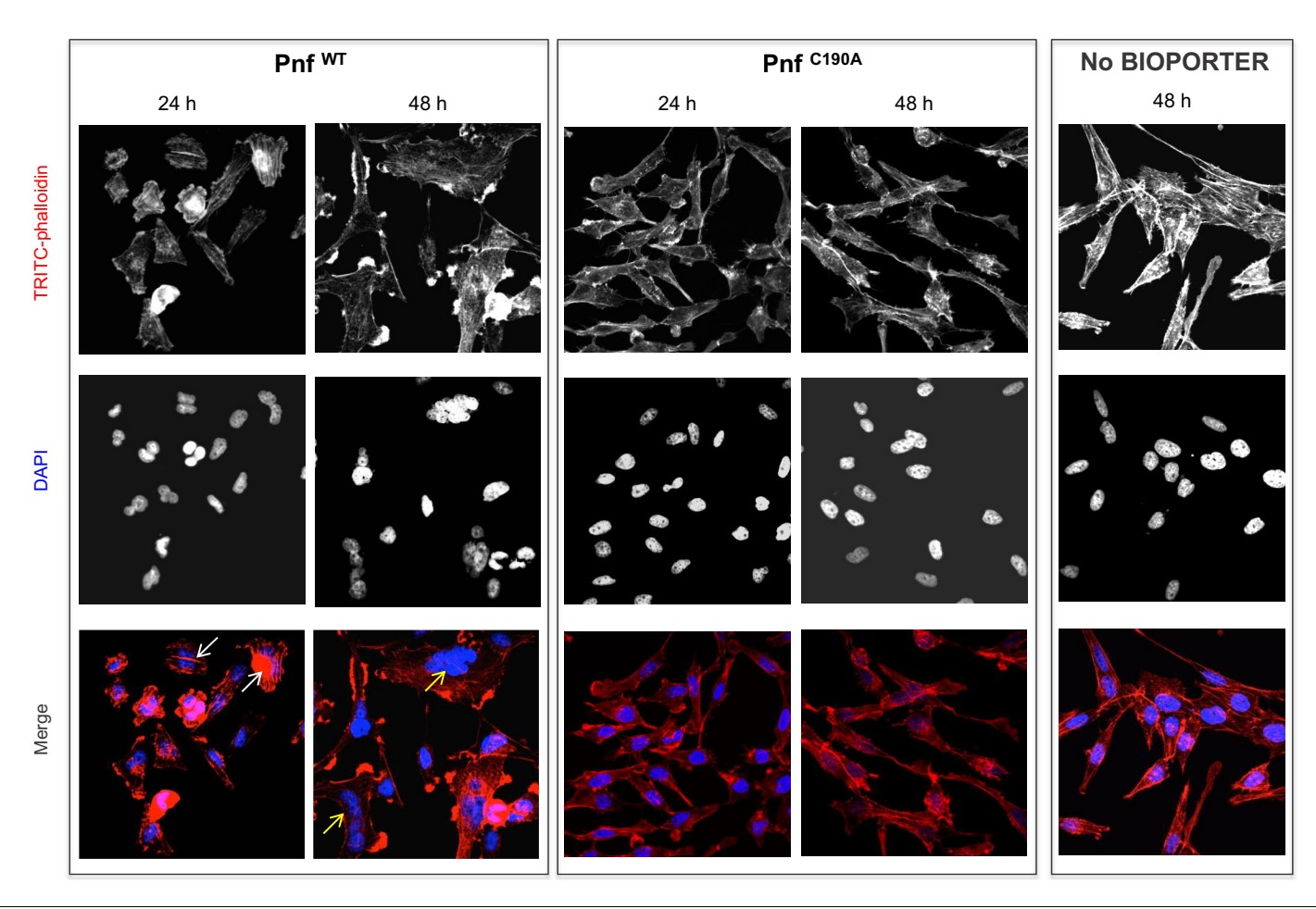

**Figure 6.** Pnf needs to gain access to the host cell cytoplasm to induce F-actin formation and multi-nucleation in cultured HeLa cells. Wild-type and inactive toxoid mutant Pnf protein was delivered topically using BioPORTER. Cell cytoskeleton is stained with TRITC-Phalloidin and the cell nuclei with DAPI. This gave rise to phenotypes consistent with the molecular targets and that of the *Yersinia* CNF2 protein homologue. Note we see the F-actin formation by 24 hr (white arrows) preceding extensive multi-nucleation of the host cell by 48 hr (yellow arrows). Note neither application of the Pnf toxoid in a BioPORTER formulation or the purified wild-type Pnf protein without BioPORTER had any observable effect on the cells.
DOI: https://doi.org/10.7554/eLife.46259.017

The following figure supplement is available for figure 6:

**Figure supplement 1.** Ab initio structural prediction of the Pnf toxin and toxoid derivative.
DOI: https://doi.org/10.7554/eLife.46259.018

vitro transglutamination and deamidation effects of purified heterologously produced Pnf on puri-fied small GTPases RhoA, Rac1 and Cdc42. The RhoA, Rac1 and Cdc42 were purified from *E. coli* heterologous expression strains using GSTrap HP affinity columns with the FPLC AKTA system in accordance with previously published protocols, as described in the methods. SDS-PAGE analysis was used to confirm good levels of purity and lack of any significant degradation. Transglutamination is the formation of a covalent bond between a free amine group, as may be found on a lysine resi-due, and the gamma-carboxamide group of glutamine. As a result protein electrophoretic mobility of the protein is altered. Deamidation is a chemical reaction in which an amide functional group is removed from the protein, which may be detected using deamidated protein specific antibodies. These experiments demonstrated that Pnf induced transglutamination and deamidation of both RhoA and Rac1 (*Figure 7*), although unlike the reported activity of CNF2, had no effect on Cdc42. As predicted the active site toxoid mutant had no enzymatic activity on any of the three Rho GTPases confirming it was a true toxoid derivative.

## Discussion

An analysis of the different subunit proteins of PVCs shows they share several elements in common with other contractile phage-tail derived systems, including the Type VI secretion system (T6SS) (*Kapitein and Mogk, 2013*) and to a lesser extent R-type pyocins (*Taylor et al., 2018*). However, PVC-like elements are distinct in two important ways. Firstly, unlike the T6SS, they require no

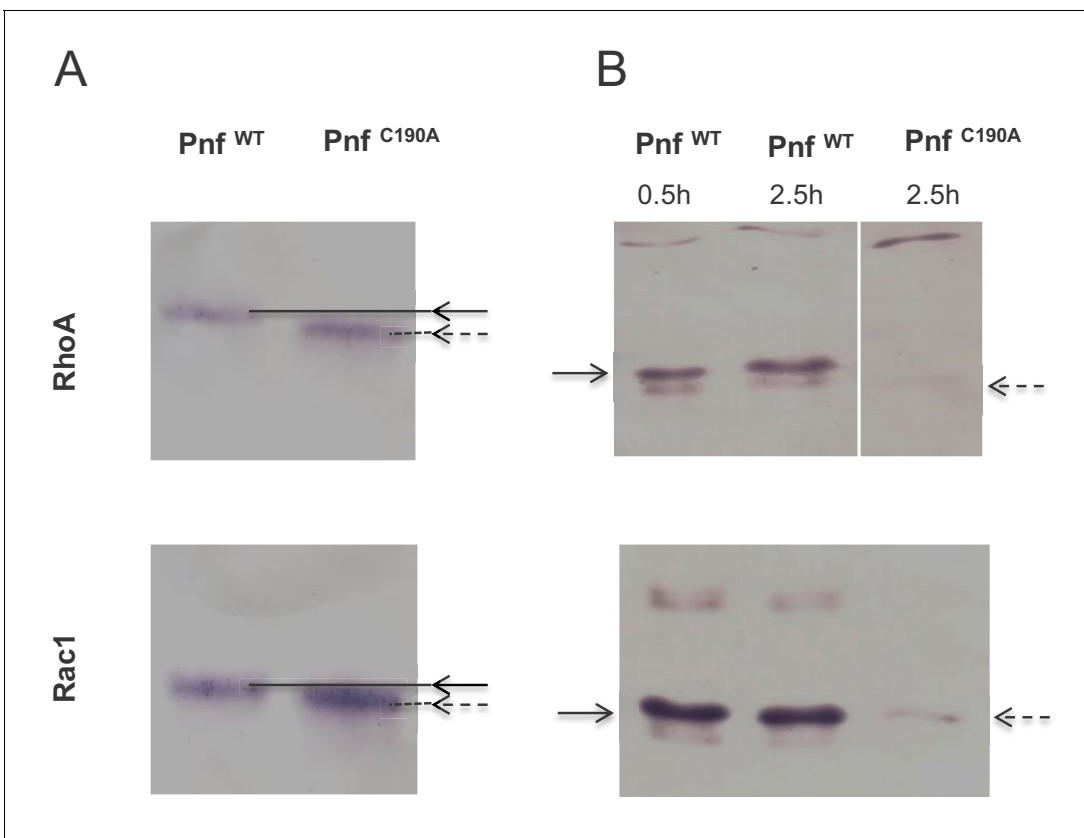

**Figure 7.** Pnf transglutaminates and deamidates purified mammalian RhoGTPAses at Gln63 (RhoA) and Gln 61 (Rac1). (**A**) For transglutamination assays a 2:1 molar ratio of small Rho GTPase to purified Pnf was incubated in transglutamination buffer in the presence of ethylediamine for 1 hr at 37˚C. Note transglutaminated GTPase runs slightly slower on the gel as visualised using anti RhoA and Rac1 antibodies. (**B**) For deamidation assays a 20:1 molar ratio of Rho GTPases; RhoA and Rac1, to purified Pnf toxin was incubated in deamidation buffer for either 30 min or 2.5 hr at 37˚C. Note deamidation is detected using an antibody specific towards deamidated Rho GTPase antigens. In both cases, the wild-type Pnf protein was active (solid arrows) while a site directed amino acid C190A toxoid mutant (in the predicted Pnf active site) showed no activity (dotted arrows).
DOI: https://doi.org/10.7554/eLife.46259.019

membrane complex for anchoring and synthesis and are freely released from the producing bacterial cell. Therefore, in common with R-type pyocins, they can act at a distance. Secondly, like T6SS but unlike R-type Pyocins, they are evolved to inject bioactive protein effectors into other cells. We hypothesise that the PVCs are evolved to specifically target eukaryotic cells, unlike T6SS, which have been shown to be able to deliver to both eukaryotes and prokaryotic competitors. However, while our previous attempts to show PVC*pnf* attachment to a range of bacterial species from different genera showed no binding we could detect (data not shown), we cannot rule out the possibility that homologues exist which are able to target prokaryotes.

We speculate that these large protein complexes are costly for the cell to produce, consistent with the observation of population heterogeneity of *pvc*-operon expression. Indeed, uncontrolled heterologous over expression of the *Pa*^ATCC43949 PVC*lopT* and PVC*pnf* operons in cosmid clones in *E. coli*, results in deletion of regions of the *pvc*-operon (confirmed by sequencing) and significant loss of culture viability (not shown). It should be noted that in a natural insect infection the vast majority of the *Photorhabdus* bacterial population are sacrificial. The majority of the population act as a food source for the replicating nematodes, with very few cells passing into the next generation of infective juvenile nematodes (*Ciche et al., 2008*). As such the population may restrict PVC production to a limited number of sacrificial cells. We have noted the presence of holin-lysin gene homologues tightly linked to certain *pvc*-operons, including both 5' and 3' to the *Pa*^ATCC43949 PVC*pnf* operon which we focus on in this publication. Release by cell lysis would be consistent with the proposed method of release of the related MAC arrays of *Pseudoalteromonas* (*Shikuma et al., 2014*). Nevertheless, to date we have not directly observed any cell lysis associated with *pvc* expression in *Photorhabdus* itself.

The finding that *Pa*^ATCC43949 PVC*pnf,* and seven other *pvc*-operons from *Pa*^PB68 and *Pl*^TT01 (not shown) all show population heterogeneity in expression, at least in vitro, suggests that they are likely deployed in a highly regulated and conservative manner. While it is difficult to fully characterise this heterogeneity in vivo, the PVC*pnf* GFP reporter strain did show restricted expression to one specific tissue, the insect trachea, and not throughout the body of the animal. In regards to these experiments, it should be noted that we did not see any melanisation response around the bacterial biomass showing GFP. The insect melanisation immune response is typically activated at sites of encapsulation. This is mediated by the recruitment of hemocytes, surrounding and enclosing foreign bodies, and entombing them in melanin. The absence of melanisation around this GFP expressing bacterial mass is consistent with the expression of anti-hemocyte virulence genes, which are likely to include the native *Pl*^TT01 *pvc*-operons.

Examination of the promoter regions has provided no clue as to the mechanism of population heterogeneity of expression. Nevertheless, the identification of RfaH and a second cryptic conserved potential operator sequence upstream of the *pvc1* genes provides a starting point for addressing this in future. Operons containing the second cryptic putative regulatory sequence include [*Pl*^TT01PVC*lopT*] and [*Pl*^TT01 PVC*u4*]. Analysis of the supplementary data from a recently published RNA-seq study (*Tobias et al., 2017*), suggests that these operons may be dependent upon Hfq/HexA activity (*Joyce and Clarke, 2003*).

Unlike many of the other genera in which we see *pvc*-like operons, *Photorhabdus* genomes encode multiple copies, typically around 5 to 6, suggesting they play important and diverse roles in the life cycle. With this in mind, we examined the conservation of the different subunit genes between operons. We observe a 'break point' in conservation, toward the 3' end of the operons. We postulate this may be due to imprecise recombination events in the 3' payload regions of *pvc*-operons, where incoming sequences, which have a GC-content that is distinct from the host genome, gradually 'erode' the upstream sequence. Alternatively, it is plausible that the lower GC at the distal end of these long operons (each of ~25 kb) may assist in strand separation during transcription, maintaining stoichiometry for these large, multi-subunit structures. Indeed low GC stretches of DNA are common origins of replication because of their reduced strand separation energy (*Meijer et al., 1979*). However, as yet we do not know whether the *pvc1* promoter serves the whole operon, or if there are additional promoters internal to the operon.

Each of the *pvc*-operons in a *Photorhabdus* genome encodes multiple paralogous copies of *pvc1/ 5* and *pvc2/3/4* genes. We were therefore surprised not to see any operons showing signs of genetic degradation. This suggests there is sufficient positive selection for maintaining these multiple operons, with each operon potentially adapted for a specific role. This hypothesis is supported by the

high variation in the Pvc13 protein sequences, which we speculate represent the host cell binding fibres. The need to maintain multiple copies of *pvc*-operons may also have arisen if the structural genes for the needle complexes are specifically adapted for delivery of their cognate *cis*-linked effector proteins in some way.

Circumstantial evidence from genomic sequences and previous work on the related AFP system of *Serratia* has suggested the needle complexes serve to deliver the *cis*-encoded effector proteins. We present here for the first time direct evidence that a linked effector protein does in fact become physically associated with the needle complex. Western blot detection of Pnf from preparations enriched for needle complexes taken from the native $Pa^{ATCC43949}$ supernatants confirmed it was being expressed in vitro and suggested it was physically associated with the complexes. In addition, physical or chemical disruption was required to release the Pnf protein for detection. When taken alongside the immuno-gold EM observations, showing Pnf could only be seen near contracted or damaged needle complexes, it confirms the protein is either sequestered inside the complex or physically associated in such as way that the TGQKPGNNEWKTGR epitope is not solvent accessible. The anti-Pvc2 antibody is able to specifically detect the protein in Western blots, however it only showed binding to what appeared to be disrupted fragments of needle complexes, again suggesting the relevant epitope is not accessible in the intact native needle complex structure. Indeed, iTasser structural model simulations of a PVC outer sheath Pvc2 protein, using the homologous *Pseudomonas* 3J9Q PDB structure of an R-type pyocin outer sheath as a model (*Ge et al., 2015*), supports this idea, suggesting the epitope is partially occluded between adjacent subunits.

While we have not yet directly demonstrated injection of Pnf into host cells by the needle complex, the results of the topical application and bioPORTER transfection experiments confirmed that the Pnf effector absolutely requires a mechanism to facilitate entry into the host cell cytoplasm to exert its effect. We argue the evidence for injection by the needle complex is very strong, and is corroborated by the SEM visualisation of needle-like structures of the correct dimensions on the surface of intoxicated hemocytes. Finally, we have confirmed that Pnf acts in a manner similar to the *Yersinia* CNF2 toxin, modifying two of the same Rho-GTPases, which correlates with the observed phenotypic effects on the cell.

## Materials and methods

### Insects, bacterial strains and growth conditions

 *Manduca sexta* (Lepidoptera: *Sphingidae*) were individually reared as described (*Reynolds et al., 1985*). Briefly, larvae were maintained individually at 25°C under a photoperiod of 17 hr light: 7 hr dark and fed on an artificial diet based on wheat germ. Larvae 1 day after ecdysis to the 5th instar were used for all experiments. Batches of wax moth larvae (75 g; Livefood UK Ltd, Rooks Bridge, UK) in their final instar stage were stored in the dark at 4°C and used within a week of receipt. DH5α *E. coli* (containing various plasmid constructs) were grown on LB agar at 37°C or in LB liquid, shaking at 200 rpm. Spontaneous rifampicin-resistant mutants of *Photorhabdus asymbiotica subsp. asymbiotica* Thai (strain PB68.1) (*Thanwisai et al., 2012*) and *Photorhabdus luminescens subsp. laumondii* TTO1 (*Duchaud et al., 2003*) were used in these studies as hosts for reporter plasmids. *Photorhabdus* were routinely cultured in LB broth or on LB agar supplemented with 0.1% (w/v) pyruvate at 30°C or 37°C (for *P. asymbiotica*). When required antibiotics were added at the following concentrations: ampicillin (Amp): 100 μg ml$^{-1}$, kanamycin (Km): 25 μg ml$^{-1}$, chloramphenicol (Cm): 25 μg ml$^{-1}$, rifampicin (Rif): 25 μg ml$^{-1}$. Human adenocarcinoma HeLa ATCC CCL2 cells (obtained from ATCC culture collection – mycoplasma free) were cultured for 10 passages in Dulbecco's modified Eagle medium (Sigma-Aldrich) containing 4.5 g/L glucose (Sigma-Aldrich), 10% heat-inactivated fetal bovine serum (Sigma-Aldrich), 2 mM glutamine (Sigma-Aldrich), 100 μg /mL penicillin, and 100 μg/mL streptomycin (Sigma-Aldrich) and incubated at 37°C and 5% $CO_2$.

### PVC gene reporter plasmid construction

Translational fusions with the *gfpmut*2 gene were constructed by PCR in a pACYC184 vector containing the *gfpmut*2 (pACYC-GFP) (*Jacobi et al., 1998*) as follows. The *pvc1*, *pnf* and *rpsM* genes (consisting of promoter regions and the first 150 bp of coding sequence) were amplified from *P. asymbiotica* $^{ATCC43949}$ genomic DNA and cloned into pACYC-*gfp* to generate pACYC-*afp1-gfp*,

pACYC-*pnf-gfp* and pACYC-*rpsM-gfp*. The constructs were further digested to release the *pvc1*, *pnf* or *rpsM* genes in frame with *gfp* and the fusion fragments were cloned into pBBR1-MCS (*Kanter-Smoler et al., 1994*) to generate pBBR1-*pvc1-gfp*, pBBR1-*pnf-gfp* and pBBR1-*rpsM-gfp*. Mating experiments were performed as previously described (*Brillard et al., 2002*) to transfer plasmid constructs into *P. luminescens* $^{TTO1}$ resulting in *Pl*$^{TTO1}$-*pvc1-gfp*, *Pl*$^{TTO1}$-*pnf-gfp* and *Pl*$^{TTO1}$-*rpsM-gfp*. Plasmid stability was confirmed in bacteria harbouring the various constructs isolated after in vivo passages. For the expanded panel of *gfp*-reporter fusions, the promoter regions for the operons selected, inclusive of the putative RfaH operator sites (if present), and the native RBS and first codon of the *pvc1* gene (approximately 500 bp upstream), were cloned in to the pAGAG vector. pAGAG was derivatised from the promoterless pGAG1 *gfp* bearing plasmid., In brief, pGAG1 was used as a template to amplify *gfpmut3\** without a start codon using primers pG_GFPfor (5'-aatgtcgaccgtaaag-gagaagaactttttc) and pG_GFPrev (5'-aatactagtggatctatttgtatagttcatccatg). The resulting product and the pGAG1 vector were cut by digestion with SalI-HF and SpeI and ligated together, thus replacing the original intact *gfpmut3\** gene with one that lacks a ribosome binding site and the first ATG codon. This reporter plasmid backbone was used to construct a panel of specific transcription-translation reporter plasmids for different *pvc*-operon 5' promotor regions. In brief regions 5' to the *pvc1* open reading frames from each operon were PCR amplified from *P. luminescens* strain TT01; PVCunit1, PVCunit4, PVClopT and PVCcif and from *P. asymbiotica* strain PB68.1; PVCPnf, PVCunit, PVClopT and PVCcif. Primers used are shown in Supplementary file 'Reporter construct primers'. All upstream promoter regions were incorporated between the KpnI and BamHI sites of the pAGAG vector. After initial cloning into *E. coli*, correct constructs, as confirmed by DNA sequencing, were transformed into the relevant *Photorhabdus* strain (*P. luminescens* TT01 or *P. asymbiotica* PB68.1) for fluorescent microscopy studies. (*Figure 2—figure supplements 1*, *2*, *3* and *4*).

## Image analysis of reporter strain expression heterogeneity

Quantitative image analysis of micrographs of *P. asymbiotica* PB68.1 harbouring the *Pa*$^{PB68}$ PVC*pnf pvc1* promoter fusion construct was performed using Fiji. Brightfield images were used to automatically detect the bacteria by converting the image to binary, followed by edge detection and particle analysis with the size of the particles set to 0.5–7 $\mu m^2$. The resulting regions of interest when then used to measure the intensity in the corresponding green channel (for GFP detection). At least four images were used per time point with a minimum total number of cells of 450 for each time point. The control consists of images taken of *P. asymbiotica* PB68.1 harbouring the negative control plasmid pAGAG and for the purposes of this analysis the mean intensity per cell calculated for each time point were combined to find a threshold value for subtraction of background fluorescence. For the calculation of the percentage number of green cells for each time point, a threshold of 1200 was used above which there were no control cells with this mean intensity value. Raw data can be seen in *Figure 2—figure supplement 2—source data 1*.

## Fluorescent reporter strain assays

### In vitro experiments

Reporter strains were cultured with shaking aeration in LB liquid medium supplemented with 20% (v/v) freshly harvested 5$^{th}$ instar *M. sexta* clarified hemolymph. To obtain the hemolymph, insects were chilled on ice for 20 min before being bled (by cutting the tip of the tail horn) into a tube on ice, containing 10 $\mu$l of saturated Phenol Thio Urea (PTU) solution, which prevents melanisation. Hemolymph was clarified by centrifugation to remove hemocytes and other debris. Bacteria were grown to late stationary phase, before microscopic visualisation using a Leica inverted epi-fluorescent microscope. In vivo experiments, we injected ca. 100 cells of the reporter strains into cohorts of 5$^{th}$ instar *M. sexta*, and allowed the infection to establish before macroscopic examination of insect tissues using a (fluorescence) dissecting microscope. We also took hemolymph samples from these insects and performed microscopic examination of fixed ex vivo hemocytes stained with phalloidin conjugate and confocal microscopy to visualise host cell cytoskeleton and any GFP expression from the recombinant bacteria. Images were acquired with a LSM510 confocal microscope (Leica).

## PVC purification from *E. coli* cosmid clone supernatants and electron microscopy

Cosmid libraries of *P. asymbiotica*[ATCC43949] were prepared in *E. coli* EC100 and arrayed into 96-well microtiter plates by MWG Biotech, Munich, Germany, as described previously (*Waterfield et al., 2009b*; *Daborn et al., 2002*). A 250 ml overnight culture of *E. coli* with the *Pa*[ATCC43949] PVC*pnf* cosmid (c4DF10) and a wild-type *Pa*[ATCC43949] were grown in LB medium supplemented with 100 µg ml$^{-1}$ ampicillin (in case of the cosmid strain) at 28°C with aeration in the dark. The cultures were centrifuged at 6800 x *g* at 4°C for 30 min at 4°C. The supernatants were decanted to remove each cell pellet, and the centrifugation procedure was repeated to remove any remaining cells. Cell-free supernatants were then centrifuged, in small batches, at 150,000 × *g* for 90 min at 4°C to harvest particulate material. The particulate pellets were washed by gentle re-suspension in 1 × Phosphate Buffered Saline (PBS) before a second centrifugation at 150,000 × *g* for 90 min at 4°C to pellet the particulate material. Each pellet was further separated by DEAE-Sepharose chromatography. 10 ml of particulate material in ice-cold PBS were mixed with an equivalent volume of DEAE-Sepharose CL-6B anion exchanger (in PBS) and the preparation was incubated at room temperature for 15 min. The Sepharose resin was harvested by centrifugation (3,000 × *g*), and the supernatant was discarded. The resin was resuspended in 40 ml of ice-cold PBS and again harvested by centrifugation. This washing step was repeated another three times, and the resin was finally resuspended in 10 ml of elution buffer (0.5 M NaCl, 50 mM phosphate buffer [pH 7.4]). The resin was removed by centrifugation, and the supernatant containing the PVCs was again centrifuged at 150,000 × *g* for 90 min at 4°C to pellet the particulate material and concentrate the needle structures in 500 µl of ice-cold PBS. For transmission electron microscopy (TEM) pioloform-covered 300-mesh copper grids that were coated with a fine layer of carbon were used as substrates for the protein fractions. The following four aqueous negative stains were tested with the protein samples: 1% uranyl acetate, 3% ammonium molybdate, 3% methylamine tungstate, and 2% sodium silicotungstate. The preferred stain, 3% methylamine tungstate, produced acceptable contrast and minimum artefacts and was subsequently used for all samples viewed by TEM. The coated grids were exposed to UV light for 16 hr immediately prior to use to ensure adequate wetting of the substrate. A 10 µl drop was applied to the TEM grid, and the protein was allowed to settle for 5 min. Liquid was absorbed with filter paper from the edge of the grid and replaced immediately with 10 µl of filtered negative stain. The drop was partially removed with filter paper, and the grids were allowed to air dry thoroughly before they were viewed with a JEOL 1200EX transmission electron microscope (JEOL, Tokyo, Japan) operating at 80 kV.

Production and isolation of secreted Pnf from wild-type *P. asymbiotica*[ATCC43949] was performed at the same time as the production and isolation of wild-type needle complex in a duplicate culture. Bacterial cultures were grown from the same starter culture and under the same conditions. However, for isolation of secreted proteins, the supernatant was filtered through 0.45 µm and 0.22 µm filters to remove any remaining bacteria and particulate matter. Then it was concentrated using 4% TCA overnight at 4°C and the next morning washed and resuspended in approximately 10 ml of 50 mM Tris-HCl pH 7.4. The secreted proteins were further concentrated using Amicon filters with NMWL of 10 kDa to 500 µl.

## Pnf cloning and heterologous expression for *Galleria* injection and antibody specificity test

*Pnf* gene was amplified from *P. asymbiotica*[ATCC43949] genomic DNA (using primers Pnf_NdeI 5'-ATATAT<u>CATATG</u>ATGTTAAAATATGCTAATCCT-3', Pnf_BamHI 5'-ATATAT<u>GGATCC</u>TTATAA-CAACCGTTTTTTAAG-3') and the PCR product was purified and cloned in-frame with a His-tag into the IPTG-inducible expression plasmid pET-15b (Novagen) to create construct pET15b-Pnf. The clone was verified by sequencing and transformed into Arctic Express competent cells (Agilent) for protein expression. A site-directed mutant of Pnf (toxoid) was generated with the QuikChange site-directed mutagenesis kit (Agilent). To construct the Pnf mutant plasmid pET15b-Pnf$_{C190A}$, pET15b-Pnf was amplified with FPLC-purified primers designed to generate a Cys to Ala substitution at position 190 (Pnf$_{C190A}$_for 5'-TCACCGAATATACCATAGTAGCACCGCTCAATGCTCCAGAC-3', Pnf$_{C190A}$_rev 5'-GTCTGGAGCATTGAGCGGTGCTACTATGGTATATTCGGTGA-3') using the following thermal profile (step 1: 95°C for 30 s, step 2: 95°C for 30 s, 55°C for 60 s, 68°C for 6 min 45 s for

16 cycles). The identity of six different positive clones was confirmed by sequencing. Subsequently, −80°C glycerol stocks were used to inoculate 5 ml of fresh LB medium supplemented with 0.2% (w/v) glucose and 100 µgml$^{-1}$ ampicillin. Bacteria were grown overnight at 30°C with shaking, and 1 ml of the culture was then harvested, re-suspended in 100 ml of the same medium, and incubated in an orbital incubator at 37°C until the optical density at 600 nm was 0.7 to 0.9. Cells were then harvested at room temperature by centrifugation at 4,000 rpm for 10 min. The pellet was re-suspended in 100 ml of fresh LB medium supplemented with the 100 µg ml$^{-1}$ ampicillin and 0.1 mM of the inducer iso-propyl-β-d-thiogalactopyranoside (IPTG). Optimized times for inductions were determined experimentally, and cells were then harvested. The bacterial cell pellet was re-suspended in 10 ml of 1x PBS and sonicated (four 20 s sonications at 45 mA using a Branson 450 digital Sonifier fitted with a tapered probe). The freshly sonicated samples were then diluted in 1x PBS for injection into *Galleria* larvae and for SDS-polyacrylamide gel electrophoresis analysis to confirm expression of the target protein. For toxicity testing cohorts of *Galleria* larvae (n = 20) were chilled on ice before injection with 10 µl of a dilution series (in sterile PBS) of sonicated cells expressing Pnf or vector control. Insects were then returned to room temperature and observed for 5 days or mortality or morbidity.

## Recombinant Pnf and small Rho-GTPase purification

ArcticExpress containing pET-15b-Pnf were initially grown in LB broth supplemented with 100 µg ml$^{-1}$ ampicillin at 37°C until OD 0.6 when Pnf expression was induced with a final concentration of 0.1 mM of IPTG at 12°C for 16 hr to produce soluble Pnf. Pnf was purified over HisTrap Ni$^{2+}$-affinity column with the fast phase liquid chromatography (FPLC) AKTA system as per the manufacturer's protocol (GE Healthcare). Plasmids pGEX-2T-wtRhoA, pGEX-2T-wtRac1 and pGEX-2T-G25K (Cdc42) were gifts from Prof Alan Hall (University College London, London, UK) and were maintained in *E. coli* DH5α grown on LB agar or in LB broth supplemented with 50 µg ml$^{-1}$ ampicillin. RhoA, Rac1 and Cdc42 were purified over GSTrap HP affinity columns with the FPLC AKTA system as per the manufacturer's protocol (GE Healthcare).

## BioPORTER assay and actin stress fibre analysis

For BioPORTER assays, 80 µl of purified wild type and mutant Pnf proteins (500 µg ml$^{-1}$), or PBS as a negative control, were added to one BioPORTER tube (Genlantis) and resuspended in 920 µl of DMEM. The samples were added to Human HeLa ATCC CCL2 cells (ATCC collection) grown in 6-well plates and incubated for 4 hr. BioPORTER/protein or PBS mixes were replaced by fresh complete medium and the cells were incubated for 20–48 hr. To visualise cell morphology and actin cytoskeleton, cells were fixed for 15 min in 4% PBS-formaldehyde, permeabilized with 0.1% Triton X-100 and stained with Tetramethylrhodamine B isothiocyanate (TRITC)-phalloidin (Sigma) and DAPI dihydrochloride (Sigma). Images were acquired with a LSM510 confocal microscope (Leica).

## Deamidation and transglutamination of Rho GTPases

### Deamidation assay

Deamidation assays were done according to previously described procedures (*Schmidt et al., 1997*) with the following modifications. Briefly, a 20:1 molar ratio of GTPase (RhoA, Rac1 or Cdc42) to toxin was incubated in deamidation buffer (50 mM NaCl, 50 mM Tris-HCl pH 7.4, 5 mM MgCl$_2$, 1 mM DTT, 1 mM phenylmethanesulphonyl fluoride) for either 30 min or 2.5 hr at 37°C. Untreated RhoA served as a negative control. After toxin treatment, samples were concentrated by the addition of 10% trichloroacetic acid and stored overnight at 4°C. Precipitated proteins were pelleted, washed with acetone, air-dried and resuspended in 20 mM Tris-HCl pH 7.4. Samples were subjected to SDS-PAGE and analysed by Western blotting using either an anti-RhoA (1:1500, Santa Cruz Biotechnology), anti-Rac1 (1:5000, Upstate Biotechnology), or anti-Cdc42 (1:1000, Santa Cruz Biotechnology) monoclonal antibody or rabbit polyclonal antisera (1:2000) that had been raised against a peptide antigen specifically designed to detect modified/deamidated RhoA/Rac1/Cdc42 (*Sugai et al., 1999*), provided by Prof A. D. O'Brien, Department of Microbiology and Immunology at Uniformed Services University, Maryland, USA. Reactive proteins were detected with either the HRP-conjugated goat anti-mouse IgG (Sigma) or donkey anti-rabbit IgG (1:3000, Sigma) followed by visualization with DAB (Sigma).

## Transglutamination assay

Transglutamination assays were done as previously described (*Schmidt et al., 1999*) with several modifications. Briefly, a 2:1 molar ratio of RhoA to toxin was incubated in transglutamination buffer (50 mM Tris-HCl pH 7.4, 8 mM $CaCl_2$, 5 mM $MgCl_2$, 1 mM DTT, 1 mM EDTA) in the presence of 50 mM ethylenediamine (which raised the pH of the buffer to 9) for 10 min or 1 hr at 37°C. As a negative control, RhoA was incubated with ethylenediamine but without toxin. Samples (0.25 µg RhoA/well) were subjected to SDS-PAGE and then processed for Western blot analyses as described above. Immunoblots were probed with a mouse anti-RhoA monoclonal antibody (1:1500, Santa Cruz Biotechnology) and reactive proteins visualised with DAB after incubation with the HRP-conjugated goat anti-mouse IgG secondary antibody.

## A bioinformatic analysis of *pvc* structural operon sequences

DNA sequences for each of the 16 conserved structural loci were clustered syntenically (all *pvc1*s, all *pvc2*'s etc.). % GC content for each CDS in each syntenic position was calculated (up to 16 observations per locus), and plotted as a boxplot via ggplot2 (*Figure 1—figure supplement 2*). The average GC content across the full operon, as well as for the whole genome, were plotted as intervals in the plot background to show the PVC loci %GC in contrast. The breakpoint was defined by use of the 'cumSEG' package in R (*Muggeo and Adelfio, 2011*). Amino acid similarity scores (*Figure 1—figure supplement 3*) were generated by CLUSTAL Omega (*Sievers et al., 2011*) multiple sequence alignment, using default parameters. Resulting pairwise alignment scores were plotted as boxplots using ggplot.

## RNA purification and RT-PCR

For in vitro transcription analysis, overnight cultures of *P. asymbiotica* were sub-cultured into liquid LB medium and grown with aeration at 28°C or 37°C 200 rpm in the dark. Planktonic cultures were collected at 4, 8 and 24 hr and mixed with a double volume of RNAlater (Ambion) and after 5 min incubation, bacteria were harvested by centrifugation and the pellets stored at −80°C. For in vivo transcription analysis, overnight cultures of *P. asymbiotica* were extensively washed in PBS and diluted in Grace's insect media (GIM) to achieve 1000 bacteria per 50 µl of culture. Each *M. sexta* larvae was injected with 50 µl of *P. asymbiotica* culture and they were placed in a humid temperature-controlled room at 28°C. After 3 hr or 6 hr of incubation, insects were bled in equal volume of GIM containing 20 mM phenylthiocarbamide (PTC). The sample was initially fractionated into plasma and total hemocytes by centrifugation at 200 x *g* at 4°C for 5 min, and plasma was further centrifuged at 6800 x *g* at 4°C for 5 min to form a bacterial pellet. For each condition, total RNA was extracted using the RNeasy Mini Kit (Qiagen) and 2 µg total RNA was treated with TURBO DNA-free Kit (Ambion) and subjected to RT-PCR using the Qiagen OneStep RT-PCR kit. Each RT-PCR reaction performed in a volume of 50 µl (containing 100 ng template RNA, 1x QIAGEN OneStep RT-PCR buffer, 400 µM dNTPs, 0.6 µM gene specific primers, 5U RNase inhibitor and 2 µl of QIAGEN One-Step RT-PCR enzyme mix) for 28 cycles.

## Acknowledgements

This work would not have been possible without the much appreciated funding by BBSRC grants BB/C008367/1 and BB/E021328/1, The Leverhulme Trust grant RPG-2015–194, EPSRC (MOAC) DTP EP/F500378/1, MRC DTP in Interdisciplinary Biomedical Research MR/N014294/1 and the Warwick University Medical School. We would also like to acknowledge Stefan Bagby lab for their help on Pnf protein purification and Chris Apark for maintaining and supplying the *Manduca sexta* insects from the University or Bath colony.

## Additional information

### Funding

| Funder | Grant reference number | Author |
|---|---|---|
| Biotechnology and Biological Sciences Research Council | BB/C008367/1 | Isabella Vlisidou<br>Nicholas R Waterfield |
| Leverhulme Trust | RPG-2015-194 | Alexia Hapeshi<br>Nicholas R Waterfield |
| Warwick Medical School | Start up package | Alexia Hapeshi<br>Nicholas R Waterfield |
| Engineering and Physical Sciences Research Council | DTPEP/F500378/1 | Joseph RJ Healey |
| Medical Research Council | MR/N014294/1 | Katie Smart |
| Biotechnology and Biological Sciences Research Council | BB/E021328/1 | Nicholas R Waterfield |

The funders had no role in study design, data collection and interpretation, or the decision to submit the work for publication.

### Author contributions

Isabella Vlisidou, Alexia Hapeshi, Joseph RJ Healey, Katie Smart, Guowei Yang, Investigation, Methodology; Nicholas R Waterfield, Conceptualization, Formal analysis, Supervision, Investigation, Writing—original draft, Project administration, Writing—review and editing

### Author ORCIDs

Alexia Hapeshi (iD) http://orcid.org/0000-0002-5717-0532
Joseph RJ Healey (iD) https://orcid.org/0000-0002-9569-6738
Katie Smart (iD) https://orcid.org/0000-0003-1748-661X
Guowei Yang (iD) https://orcid.org/0000-0002-3090-6677
Nicholas R Waterfield (iD) https://orcid.org/0000-0002-8044-7763

### Decision letter and Author response

Decision letter https://doi.org/10.7554/eLife.46259.025
Author response https://doi.org/10.7554/eLife.46259.026

## Additional files

### Supplementary files

• Supplementary file 1. Table of PCR primers. Oligonuclotides used for the construction of pAGAG based reporter plasmids used in *Figure 2—figure supplement 2*, *3* and *4*.
DOI: https://doi.org/10.7554/eLife.46259.020

• Transparent reporting form
DOI: https://doi.org/10.7554/eLife.46259.021

### Data availability

Photorhabdus asymbiotica ATCC43949 complete genome. GenBank Accession FM162591 (https://www.ncbi.nlm.nih.gov/nuccore/FM162591).

The following previously published dataset was used:

| Author(s) | Year | Dataset title | Dataset URL | Database and Identifier |
|---|---|---|---|---|
| Wilkinson P, Waterfield NR, Crossman L, Corton C Sanchez-Con- | 2009 | Photorhabdus asymbiotica ATCC43949 complete genome | https://www.ncbi.nlm.nih.gov/nuccore/FM162591.1 | GenBank, FM162591 |

treras M, Vlisidou I, Barron A, Bignell A, Clark L, Ormond D, Mayho M, Bason N, Smith F, Simmonds M, Churcher C, Harris D, Thompson NR, Quail M, Parkhill J, Ffrench-Constant RH

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
