## [Decision Letter]

Thank you for submitting your article "*Photorhabdus* Virulence Cassettes:

extracellular multi-protein needle complexes for delivery of small protein effectors into host cells" for consideration by *eLife*. Your article has been reviewed by Gisela Storz as the Senior Editor, a Guest Reviewing Editor, and three reviewers. The reviewers have opted to remain anonymous.

The reviewers have discussed the reviews with one another and the Reviewing Editor has drafted this decision to help you prepare a revised submission.

Summary:

The manuscript describes the packing of the toxin that insect pathogenic bacteria *Photorhabdus* use as a virulence factor into a nanosyringe. *Photorhabdus* Virulence Cassettes (PVC) are evolutionarily related to the Serratia anti-feeding prophage, the *Pseudoalteromonas* MAC system and the bacterial type 6 secretion system. The authors' computational analysis revealed three classes of PVC systems and identified genes homologous to the MAC gene cluster and to *Burkholderia mallei* T6SS, and they identified putative effectors for the three classes of PVC. One of the effectors, Pnf, was shown to directly associate with the PVC tube, potentially allowing long-range effector delivery following the release of the PVC tube from the bacteria. Addition of PVC*pnf* needle complexes to insect hemocytes led to membrane ruffling and the appearance of ~150 nm long linear structures on the cell surface. The authors further demonstrate that the effector Pnf – which was previously shown to interfere with actin cytoskeleton of the host cells – acts by modifying small Rho GTPases RhoA and Rac1 by transglutamination and deamidation.

Overall the manuscript is interesting and advances the understanding of the PVC systems, their relation to the R-type pyocins, MAC and T6SS systems, and the mechanisms of effector action. The data is of high quality and the manuscript is mostly well written. However, some parts of the manuscript are rather convoluted, and several important revisions need to be made before the manuscript can be published. We would welcome a revised manuscript.

Essential revisions:

1) Bioinformatic analysis identified several features that were, unfortunately, not further tested or discussed in this study e.g. role of potential RfaH operator sequence. Are these elements involved in the heterogeneity reported later in the study? It might be helpful if the authors gave some background on RfaH here (rather than in the Discussion section) to aid in understanding what was done. Moreover, the section on the potential mobility of PVC genetic loci is very speculative.

2) In order to confirm the in vivo expression of the PVC loci, bacteria containing GFP reporter constructs were injected into Manduca larvae before samples were taken for microscopy. As the *gfpmut2* allele used here encodes a stable derivative of GFP, it is important to know the level of GFP expression in the inoculum and also how long post-injection before samples were taken. Did the bacteria grow in the insect to allow potential dilution of GFP already present in the bacteria? Alternatively, it is possible to use a *gfp* allele encoding an unstable protein e.g. GFP(LVA). The RT-PCR data presented in this supplementary figure are effective in supporting in vivo expression of this locus and should be used as a main text figure.

3) The reviewers think that more elaborate control of protein quality should be presented in the manuscript. In particular, GFP fusions of the PVC locus from *P. asymbiotica* is expressed in vitro and in vivo was monitored in *P. luminescens*. Are the same regulatory circuits present in *P. luminescens*? For example, the authors do not report any heterogeneity of expression in this heterologous system.

4) The in vitro experiments in the Results section have not been described rigorously enough and they are hard to follow, particularly in subsection “The Pnf effector protein is physically associated with the PVC needle complex”. Was the total protein concentration in the clarified supernatant sample similar to the concentration in the particle preps sample? The statement about the need for chemical or physical disruption of the needle for detection of Pnf needs to be more specific.

5) The section on the effects of heterologously expressed and purified Pnf on the actin cytoskeleton is missing important controls. The proteins Pnf and Pnf_C190A_ must be folded and of sufficient purity for the experiments to be meaningful. Protein purity and quality could be assessed by SDS-PAGE or anion exchange and/or size exclusion chromatography or CD analysis. It would be interesting to know more about secondary structure content, regions of potential disorder or Pnf, potential membrane association and the motivation for the particular design of the C190A mutant. Please comment on purity and quality of Rho1 and Rac1.

6) The findings of the manuscript should be discussed with respect to the recently published structure of the PVC (Jiang et al., 2019). Particularly, Figure 1—figure supplement 3 and Figure 1—figure supplement 4 and subsection “The Pnf effector protein is physically associated with the PVC needle complex” should be updated.

7) The authors could elaborate on the possible mechanism of the release of PVC from the bacterial cells beyond the Discussion section? Are there any toxic effects on the bacteria and are there any potential immunity proteins which potentially could be identified by bioinformatic analysis?

8) Subsection “A bioinformatic analysis of *pvc* structural operon sequences”: the authors' own phylogenetic analyses are not shown (although relevant analyses are referred to subsequently). Please present the results or delete reference to authors' unpublished studies.

9) Figure 1 legend: Not clear if the figure refers to *pvc16* as class II or III? Please clarify this section.

10) Figure 6 legend: this says there is no activity of the mutant but in the Rac1 image, there is a faint band. "No Pnf" controls would be valuable in these figures.

11) Discussion section: expression in spiracles is mentioned here but was not listed in the Results.

12) Subsection “Deamidation and Transglutamination of Rho GTPases”: the pH is unclear here since the buffer (7.4) and the ethylene diamine (9.0) are both listed but this appears to be the same solution.

13) The arguments expressed in the Discussion section strongly support that PVCs are more similar to R-type pyocins than to the T6SS. The most important distinction is that there is no membrane complex in the PVC. PVCs have more or a less uniform length and act on the target at a distance, therefore the statement that PVC is less similar to R-type pyocins than to T6SS should be revised.

14) Please note the *eLife* guidelines for titles: Two-part titles and/or punctuation are to be avoided: “In the interests of style and clarity, the titles of *eLife* research papers should not use colons, dashes, exclamation marks, or brackets (unless needed for scientific reasons). Two-part titles should also be avoided (though some exceptions can be made for Tools and Resources).” Furthermore, authors should avoid acronyms in the title: “Titles of *eLife* research papers should avoid unfamiliar abbreviations or acronyms, or authors should spell out in full or provide a brief explanation for any acronyms.” Please revise your title with this advice in mind. A possibility: "The PVC element of *Photorhabdus asymbiotica* virulence cassettes deliver protein effectors directly into target eukaryotic cells".

[Editors' note: further revisions were requested prior to acceptance, as described below.]

Thank you for submitting the revised version of the manuscript "The PVC element of *Photorhabdus asymbiotica* virulence cassettes deliver protein effectors directly into target eukaryotic cells". It has been favourably evaluated by the reviewing editor. The updated version has adequately addressed most of the original concerns of the reviewers; however, a few points still need to be dealt with. The remaining revisions will not affect the conclusions of the manuscript, but some of them will serve as important controls, and others are clarifications in the text to make the manuscript more accessible to the general readers of *eLife*.

In particular, reviewer's comment 3 has not been fully addressed. If the authors have the data to support similar heterogeneity levels in the two used strains these should be added and briefly described.

Comment 5: please add the considerations about protein quality of Pnf and Pnf C190A to the text of the manuscript. Please comment on purity and quality of Rho1 and Rac1. We believe that the goal of the question about the secondary structure content has not to suggest that you embark on structure determination, but the computational prediction of secondary structure and a comment on the potential ordered and disordered parts of the protein.

Updated subsection “The Pnf effector protein is physically associated with the PVC needle complex” make statements about expression levels observed by florescent microscopy and refer to "not shown" data. Please present this data in the supplementary material and provide quantification for the percentage of green cells.

---

## [Author Response]

Essential revisions:1) Bioinformatic analysis identified several features that were, unfortunately, not further tested or discussed in this study e.g. role of potential RfaH operator sequence. Are these elements involved in the heterogeneity reported later in the study? It might be helpful if the authors gave some background on RfaH here (rather than in the Discussion section) to aid in understanding what was done. Moreover, the section on the potential mobility of PVC genetic loci is very speculative.

We appreciate the reviewer’s comments regarding the untested role of RfaH in the regulation of the PVC operons. In attempts to address this, we did in fact try to knock-out the *Photorhabdus luminescens* TT01 *rfaH* gene, using a recombineering strategy, that normally works in our hands for this strain. However, we were not able to delete the gene, despite several attempts, suggesting an essential role in *Photorhabdus*. We also procured an *E. coli rfaH* Keio collection knock out strain, but its unusual cellular morphology and phenotypic behavior led us to believe it was unsuitable for any relevant experiments. We have now included this and a brief discussion of *rfaH* into this Results section as suggested. We accept the reviewer’s comment regarding the PVC operon motility and have reduced and simplified this text.

2) In order to confirm the in vivo expression of the PVC loci, bacteria containing GFP reporter constructs were injected into Manduca larvae before samples were taken for microscopy. As the gfpmut2 allele used here encodes a stable derivative of GFP, it is important to know the level of GFP expression in the inoculum and also how long post-injection before samples were taken. Did the bacteria grow in the insect to allow potential dilution of GFP already present in the bacteria? Alternatively, it is possible to use a gfp allele encoding an unstable protein e.g. GFP(LVA). The RT-PCR data presented in this supplementary figure are effective in supporting in vivo expression of this locus and should be used as a main text figure.

In this case, in vitro reporter expression (in LB medium) is only seen in a sub-population of cells, as illustrated by similar constructs shown in Figure 2—figure supplement 1. It should be noted that the blood samples and dissections were performed at 12 hours post infection, after injection of a small inoculum of only 100 CFU as described in the Materials and methods section. This indicates that the bacteria would necessarily have been replicating up to the time of examination (in order to see them at all), greatly increasing their numbers and initiating a normal infection. Thus, in answer to the reviewer’s question, the green cells are highly unlikely to represent those initially injected. The microcolonies seen in Figure 2B and bacterial cells in 2A therefore represent the activity of cells undergoing an active infection cycle. We have included text to highlight this.

As suggested by the reviewer, we will move the RT-PCR data shown into a figure in the main text (Figure 2C) to further support our findings relating to in vivo expression.

3) The reviewers think that more elaborate control of protein quality should be presented in the manuscript. In particular, GFP fusions of the PVC locus from P. asymbiotica is expressed in vitro and in vivo was monitored in P. luminescens. Are the same regulatory circuits present in P. luminescens? For example, the authors do not report any heterogeneity of expression in this heterologous system.

As the same cis-encoded potential regulatory elements are seen in homologous operons from all *Photorhabdus* strains (and indeed in other genera such as *Yersinia*) that we have looked at, we have no reason to believe that *P. asymbiotica* ATCC43949 and *P. luminescens* TT01 would have different regulatory circuits for *pvc* expression. In a previous study we compared gene expression and pheno-array activity of TT01 and ATCC43949 (see Mulley et al., 2015). We do indeed see the same heterogeneity of expression in all strains we have looked at when grown at the normal insect infection temperature of 28°C. While in Figure 2—figure supplement 1 we only show the activity of *Pa*^ATCC43949^ promotors in a closely related *P. asymbiotica* strain (*Pa*^PB68^) we also have equivalent data for these reporters in *P. luminescens* TT01 which we excluded for brevity. These could be included in the figure also if it is felt necessary.

4) The in vitro experiments in the Results section have not been described rigorously enough and they are hard to follow, particularly in subsection “The Pnf effector protein is physically associated with the PVC needle complex”. Was the total protein concentration in the clarified supernatant sample similar to the concentration in the particle preps sample? The statement about the need for chemical or physical disruption of the needle for detection of Pnf needs to be more specific.

We isolated the needle complexes (NCs) from both the wild-type *Photorhabdus asymbiotica* and *E. coli* cosmid clone (c4DF10) grown in 250 ml of LB media for 16 hrs. The particulate matter including NC was isolated from both cultures as described in Materials and methods section and the NC in both cases was resuspended in 0.5 ml of PBS. The reason we have selected the specific growth phase and culture volume relies on previous studies reporting successful detection of *Photorhabdus* secreted toxins under similar conditions. Detection of secreted Pnf from wild-type *P. asymbiotica*^ATCC43949^ was performed at the same time as the isolation of wild-type NC in a duplicate culture. Bacterial cultures were grown from the same starter culture and under the same conditions. However, for isolation of secreted proteins, the supernatant was filtered through 0.45 μm and 0.22 μm filters to remove any remaining bacteria and particulate matter. Then it was concentrated using 4% TCA overnight at 4°C and the next morning washed and resuspended in 10 ml of 50 mM Tris-HCl pH 7.4. The secreted proteins were further concentrated using Amicon filters with NMWL of 10kDa to 0.5 ml. It should be noted that we cannot really compare the two NC preparations as one relies on the natural expression levels, while the other is constitutive expression on the cosmid clone. The chemical and physical methods of needle complex are we believe self-explanatory. We have included this text in the manuscript.

5) The section on the effects of heterologously expressed and purified Pnf on the actin cytoskeleton is missing important controls. The proteins Pnf and Pnf_C190A_ must be folded and of sufficient purity for the experiments to be meaningful. Protein purity and quality could be assessed by SDS-PAGE or anion exchange and/or size exclusion chromatography or CD analysis. It would be interesting to know more about secondary structure content, regions of potential disorder or Pnf, potential membrane association and the motivation for the particular design of the C190A mutant. Please comment on purity and quality of Rho1 and Rac1.

Both the wild-type and mutant toxoid were produced and purified in the same bacterial expression host from the same constructs and under the same conditions (over-expression for 16 hours at low temperature to facilitate solubility). Both proteins were purified at similar yields and none of them appears to be degraded during purification (checked with SDS-PAGE both run at the same size). In addition, the correct folding of the Pnf WT is supported by the clear expected phenotypes obtained by the protein transfection experiments into mammalian cells. It remains formally possible that the toxoid no longer folds correctly, but we would argue that this would only be the equivalent as a heat denatured control. We suggest that determination of the actual structure of the proteins is a little over the top, and not something routinely done in analogous published studies.

6) The findings of the manuscript should be discussed with respect to the recently published structure of the PVC (Jiang et al., 2019). Particularly, Figure 1—figure supplement 3 and Figure 1—figure supplement 4 and subsection “The Pnf effector protein is physically associated with the PVC needle complex” should be updated.

We have now modified the text to include reference to this recent publication as requested. We have also improved Figure 1—figure supplement 3 and Figure 1—figure supplement 4 to include a more up to date HHPRED analysis table that is colour coded to match that used in the PVC structure publication. In addition, we have modified the text in subsection “The Pnf effector protein is physically associated with the PVC needle complex” and subsection “The Pnf protein requires delivery into the eukaryotic cell cytoplasm to exert its

Effect” to discuss the location of the Pvc2 anti peptide antibody epitope in relation to the recently published structure.

7) The authors could elaborate on the possible mechanism of the release of PVC from the bacterial cells beyond the Discussion section? Are there any toxic effects on the bacteria and are there any potential immunity proteins which potentially could be identified by bioinformatic analysis?

We have expanded the text in this section in response to the reviewer’s suggestion. In unpublished work we have data to suggest that either a restricted small sub-population can produce very large quantities of PVCs, that are released by lysis, or that the needle complex can in fact be secreted. We hope this work will form part of a subsequent publication.

We do discuss toxic effects of unregulated PVC expression in the heterologous *E. coli* cosmid expression strains in the manuscript. We do have loss of viability data in *E. coli* we could include if deemed necessary. In addition, we have tested purified PVC*pnf* needle complexes themselves against a range of different bacteria genera, and saw no toxicity, or indeed even cell surface binding using TEM studies. We mention this in the Discussion section. It does not seem useful for us to include this “negative” data however.

As such we do not really understand the question regarding immunity proteins. As mentioned above, unlike Type 6 secretion systems, the PVCs do not appear to target other bacteria. As such the toxins they deliver are apparently evolved against eukaryotic targets. Therefore, there is no assumed requirement for immunity proteins for the expression of PVCs (and their cognate toxins) in the bacteria themselves.

8) Subsection “A bioinformatic analysis of pvc structural operon sequences”: the authors' own phylogenetic analyses are not shown (although relevant analyses are referred to subsequently). Please present the results or delete reference to authors' unpublished studies.

We do not wish to increase the complexity and size of the publication further so as suggested by the reviewer we have removed the reference to our unpublished phylogenetic analysis.

9) Figure 1 legend: Not clear if the figure refers to pvc16 as class II or III? Please clarify this section.

We have edited the figure legend text to improve consistency and clarity and hope this addresses the issue.

10) Figure 6 legend: this says there is no activity of the mutant but in the Rac1 image, there is a faint band. "No Pnf" controls would be valuable in these figures.

The specificity of the anti-63 polyclonal against the exact same constructs (not only plasmids but also expression host) has been shown in McNichol et al. (2006) paper Figure 6, where we can see a faint non-specific band for both RhoA and Rac1 incubated without toxin (lane 1 on western blot panels). The only way to address this with any precision may be to use 2D electrophoresis and/or MS analysis as deamidation of glutamine alters the isoelectric point (pI) of the protein. We feel this additional effort would not really add much to the manuscript.

11) Discussion section: expression in spiracles is mentioned here but was not listed in the Results.

Apologies, this was an error and the comment in the discussion has now be changed to “insect trachea”.

12) Subsection “Deamidation and Transglutamination of Rho GTPases”: the pH is unclear here since the buffer (7.4) and the ethylene diamine (9.0) are both listed but this appears to be the same solution.

Please note that the addition of ethylenediamine increased the pH to 9.

13) The arguments expressed in the Discussion section strongly support that PVCs are more similar to R-type pyocins than to the T6SS. The most important distinction is that there is no membrane complex in the PVC. PVCs have more or a less uniform length and act on the target at a distance, therefore the statement that PVC is less similar to R-type pyocins than to T6SS should be revised.

In response to the first comment here, we believe the reviewer may have misread this as the sentence referred to actually says; “An analysis of the different subunit proteins of PVCs show they share several elements in common with other contractile phage-tail derived systems, including the Type VI secretion system (T6SS) (Leiman et al., 2010) and to a lesser extent R-type pyocins (Russell et al., 2014).”

We argue that we do highlight these differences/similarities between the PVCs and T6SS and R-type pyocins in a clear way. While the lack of membrane complex, standardised length and action at a distance are indeed more reminiscent of R-type pyocins, genetically PVCs are in many respects more like T6SS than the more phage-like pyocins. Nevertheless, we have made minor modification to this text to clarify this.

14) Please note the eLife guidelines for titles: Two-part titles and/or punctuation are to be avoided: “In the interests of style and clarity, the titles of eLife research papers should not use colons, dashes, exclamation marks, or brackets (unless needed for scientific reasons). Two-part titles should also be avoided (though some exceptions can be made for Tools and Resources).” Furthermore, authors should avoid acronyms in the title: “Titles of eLife research papers should avoid unfamiliar abbreviations or acronyms, or authors should spell out in full or provide a brief explanation for any acronyms.” Please revise your title with this advice in mind. A possibility: "The PVC element of Photorhabdus asymbiotica virulence cassettes deliver protein effectors directly into target eukaryotic cells".

We are happy to adopt the new title suggested.

[Editors' note: further revisions were requested prior to acceptance, as described below.]

In particular, reviewer's comment 3 has not been fully addressed. If the authors have the data to support similar heterogeneity levels in the two used strains these should be added and briefly described.

We have now expanded the Figure 2 supplementary material to include representative micrographs from a time course of reporter activity from four PVC operons from each of two species, *P. luminescens*^TT01^ and *P. asymbiotica*^PB68.1^ (so a total of 8 PVC operons now). These can be viewed as Figure 2—figure supplement 3 and Figure 2—figure supplement 4. We have also now added pertinent descriptive text to the manuscript and expanded the Materials and methods section to include construction of the additional seven plasmids (including a primer list as a supplementary document).

Comment 5: please add the considerations about protein quality of Pnf and Pnf C190A to the text of the manuscript. Please comment on purity and quality of Rho1 and Rac1. We believe that the goal of the question about the secondary structure content has not to suggest that you embark on structure determination, but the computational prediction of secondary structure and a comment on the potential ordered and disordered parts of the protein.

We have now added comments regarding the methods used and purity of the various protein preparations. In addition, we have performed Phyre2 computational structural predictions of the wild-type Pnf and toxoid sequences. We have added text to discuss that the single amino acid change in the Pnf active site (C190A) causes minimal overall predicted structural differences in the two proteins. A summary figure has now been included as Figure 6—figure supplement 1 to illustrate a comparison of these models.

Updated subsection “The Pnf effector protein is physically associated with the PVC needle complex” make statements about expression levels observed by florescent microscopy and refer to "not shown" data. Please present this data in a supplementary material and provide quantification for the percentage of green cells.

We have now included an image analysis (Figure 2—figure supplement 2) to objectively quantify heterogeneity of GFP expression in the reporter strain shown in Figure 2—figure supplement 1. A description is included in the text and methods included in the figure legend. We also include the raw data as an excel file.